# Effect of Steam Quality on Extra-Heavy Crude Oil Upgrading and Oil Recovery Assisted with PdO and NiO-Functionalized Al₂O₃ Nanoparticles

**Luisana Cardona** [1,†], **Oscar E. Medina** [2,†], **Santiago Céspedes** [2], **Sergio H. Lopera** [1], **Farid B. Cortés** [2] **and Camilo A. Franco** [2,*]

1   Research Group Yacimientos de Hidrocarburos, Department of Processes and Energy, Faculty of Mines, National University of Colombia, Medellín 050034, Colombia; lcardonar@unal.edu.co (L.C.); shlopera@unal.edu.co (S.H.L.)

2   Grupo de Investigación en Fenómenos de Superficie—Michael Polanyi, Department of Processes and Energy, Faculty of Mines, National University of Colombia, Medellín 050034, Colombia; oemedinae@unal.edu.co (O.E.M.); sacespedeszu@unal.edu.co (S.C.); fbcortes@unal.edu.co (F.B.C.)

*   Correspondence: caafrancoar@unal.edu.co

†   Both authors contributed the same to do the manuscript.

**Abstract:** This work focuses on evaluating the effect of the steam quality on the upgrading and recovering extra-heavy crude oil in the presence and absence of two nanofluids. The nanofluids AlNi1 and AlNi1Pd1 consist of 500 mg·L⁻¹ of alumina doped with 1.0% in mass fraction of Ni (AlNi1) and alumina doped with 1.0% in mass fraction of Ni and Pd (AlNi1Pd1), respectively, and 1000 mg·L⁻¹ of tween 80 surfactant. Displacement tests are done in different stages, including (i) basic characterization, (ii) waterflooding, (iii) steam injection at 0.5 quality, (iv) steam injection at 1.0 quality, (v) batch injection of nanofluids, and (vi) steam injection after nanofluid injection at 0.5 and 1.0 qualities. The steam injection is realized at 210 °C, the reservoir temperature is fixed at 80 °C, and pore and overburden pressure at 1.03 MPa (150 psi) and 5.51 MPa (800 psi), respectively. After the steam injection at 0.5 and 1.0 quality, oil recovery is increased 3.0% and 7.0%, respectively, regarding the waterflooding stage, and no significant upgrade in crude oil is observed. Then, during the steam injection with nanoparticles, the AlNi1 and AlNi1Pd1 increase the oil recovery by 20.0% and 13.0% at 0.5 steam quality. Meanwhile, when steam is injected at 1.0 quality for both nanoparticles evaluated, no incremental oil is produced. The crude oil is highly upgraded for the AlNi1Pd1 system, reducing oil viscosity 99%, increasing the American Petroleum Institute (API)° from 6.9° to 13.3°, and reducing asphaltene content 50% at 0.5 quality. It is expected that this work will eventually help understand the appropriate conditions in which nanoparticles should be injected in a steam injection process to improve its efficiency in terms of oil recovery and crude oil quality.

**Keywords:** adsorption; nanoparticles; steam injection; steam quality; thermal enhanced oil recovery

## 1. Introduction

The Heavy (HO) and extra-heavy oil (EHO) reservoirs are targeted by non-thermal and thermal enhanced oil recovery technologies. The non-thermal methods include miscible [1–3], and non-miscible gas injection [4–6], while thermal methods mainly use air or steam flooding [7–9]. Steam flooding is accomplished by bringing heat into the reservoir to unlock heavy oil recovery by reducing viscosity [9]. However, the technique presents some major challenges, such as steam over-ride, heat loss, and low thermal conductivities of the rocks. The steam quality (X) influences the displacement efficiency and the viscosity reduction, and other mechanisms that help determine optimum flooding conditions [10–13]. It is well known that the steam quality and temperature affect the oil recovery from the steam flooding process, reducing the oil production by more than 50% when X = 0.5 [14]. Kirmani et al. [11] developed a study in which the integration of injection temperature and steam quality upon steam flooding on oil

recovery was done. The results indicated that high temperature (245 °C) and the moderate value of steam quality (X = 0.5) give the best outcome regarding oil recovery by steam flooding in an economical way. Also, they demonstrated that high steam quality (X = 1.0) and high temperature increase the oil recovery regarding the scenario with low temperature (200 °C) and moderate quality. In other work, Zao et al. [15] evaluated the effect of steam quality on oil recovery after steam huff-and-puff for a heavy oil sample. The experimental results show that the steam injection quality has a noticeable influence on the expansion of the steam chamber. As the steam quality increases, the steam chamber formation time is shortened, the distance of the steam chamber expansion increases, and the adequate steam flooding time is prolonged, increasing the oil recovery rate. Nevertheless, under field conditions, it is not easy to achieve steam qualities higher than 80%; therefore, the performance of the technology is not as expected [13].

Some researchers have investigated different operating conditions in steam injection processes. Dong et al. [16] demonstrate that the main problems during steam injection were identified as a steam breakthrough, low sweep efficiency, and low steam efficiency. In addition, it is highlighted that the future of thermal processes lies in offshore fields. In addition, other types of technologies, such as the use of electricity, are currently under development (laboratory scale). The importance of studies that integrate Enhanced Oil Recovery (EOR) techniques at different scales and their effect through process simulation was also highlighted. Wen et al. [17] studied the interaction between heavy crude oil and water in the presence of the catalyst molybdenum oleate. This was done experimentally by replicating the aquathermolysis process in an autoclave, reducing crude oil viscosity by 90% at 240 °C.

This work demonstrated the effect of metals as catalysts to decrease the viscosity of crude oil and high temperatures, leading to higher oil production.

Recently, nanomaterials have been proposed to overcome the different challenges regarding thermal enhanced oil recovery (TEOR) of HO and EHO by decomposing heavy oil fractions like asphaltenes and resins. Some nanoparticles of different chemical nature have been employed, including $SiO_2$ [4,18–21], $CeO_2$ [22–27], $TiO_2$ [28], and $Al_2O_3$ [28,29], among others. The materials are characterized as having a high affinity for heavy oil fractions and reducing the asphaltene decomposition temperature near 300 °C. In search of improving their catalytic activity, the benefits of functionalized materials have been reported in the literature, which with low concentrations of transition metal oxides, can reduce the decomposition temperature close to 200 °C. Some of the most used active phases in literature are Ni [18,19,30,31], Pd [18,23,24], Fe [19,26,32], Co [26,33], and Au [34]. Recently, it has been demonstrated that the functionalized nanoparticles improve the conductivity of porous media, and hence, the heat transfer from the steam to the fluids.

Some authors have included these nanoparticles to assist displacement tests simulating steam injection processes. Medina et al. [25] injected a dispersed nanofluid in a steam stream, containing 0.2% in mass fraction of $CeO_2$ nanoparticles doped with 1.1% and 0.89% in mass fraction of Ni and Pd. They obtained 93.0% of oil recovery at 210 °C and X = 0.7. Franco et al. [21] used a nanofluid containing 0.05% in mass fraction of $SiO_2$ nanoparticles functionalized with 1.0% in mass fraction of Ni and Pd. The nanofluid was injected in batch at 0.5 quality. In their work, water consumption was considerably reduced. The oil recovery was increased up to 50%, and the crude oil quality was improved, reducing the asphaltene content and the oil viscosity. In another investigation, Afzal et al. [35] analyzed the catalytic effect at different concentrations of $Fe_2O_3$ nanoparticles on the viscosity of heavy oil at various temperatures. The treated crude oil had a viscosity of 16,000 cP and °API of 12. In this investigation, through displacement tests carried out on a laboratory scale, improvements in the relative viscosity of the crude oil of up to 60% were obtained considering steam injection and 0.5% wt of nanoparticles. In the displacement tests, an improvement in production from 38.31% to 68.41% was identified, due to the use of nanomaterials. Finally, the effect of the catalyst size was analyzed by Hamedi et al. [36], where nanometric nickel is compared with micrometric nickel in oil recovery

and upgrading during steam injection. As a result, it is highlighted that the nanometric particles reduce the oil viscosity from 8500 cP to 1530 cP, and oil recovery was increased by 8.0%. This work highlights the importance of the catalysts, where the larger the size, the higher the concentration of dispersant is needed to maintain stability, which affects the aquathermolysis process.

However, most of the studies reported in the literature that includes nanotechnology to assist the steam injection have been done at a fixed steam quality neither evaluated the effect of the quality. Therefore, the impact of nanoparticles during steam injection at different qualities for heavy oil recovery and upgrading is not yet clear.

In this context, the main objective of this study is to evaluate the effect of the steam quality in crude oil recovery and upgrading during steam injection in the presence of nanocatalysts. For this, alumina nanoparticles were functionalized with nickel and palladium, obtaining a monoelemental system consisting of 1.0% in mass fraction of Ni, and a bielemental with 1.0% in mass fraction of Ni and Pd. The samples were tested at static and dynamic conditions to evaluate their capacity to decompose heavy oil fractions and upgrade crude oil quality. This is the first time the steam quality effect is assessed in oil recovery and upgrading assisted by different nanocatalyst, which will eventually help understand the suitable conditions in which nanocatalysts should be injected in an upscaling process.

## 2. Materials and Methods

### 2.1. Materials

Commercial $Al_2O_3$ nanoparticles functionalized with 1.0% in mass fraction of Ni (AlNi1) and 1.0% in mass fraction of Ni and Pd (AlNi1Pd1) were provided from Petroraza S.A.S (Medellín, Antioquia, Colombia). The reservoir fluids are composed of an extra-heavy crude oil of 6.9° API, a viscosity of $2.3 \times 10^6$ cP at 25 °C, and saturates, aromatics, resins, and asphaltenes (SARA) content of 13.0%, 16.9%, 49.9%, and 20.2% in mass fraction, respectively. A synthetic brine of 2000 mg·$L^{-1}$ NaCl ($\geq$99.5%, Merck KGaA, Darmstadt, Germany) was used for the displacement test. The EHO was also used for asphaltene isolation following the protocols described elsewhere [18]. The steam was generated using deionized water (conductivity of 3 μS·$cm^{-1}$). Also, a hydrogen donor (Tetralin $C_{10}H_{12}$, Petroraza S.A.S, Medellín, Antioquia, Colombia) was used during displacement tests. Silica sand (Ottawa sand, US sieves 20–40 mesh -Minercol S.A) was used as porous media, and it was cleaned with methanol (99.8% purity, Merk KGaA, Darmstadt, Germany), toluene (99.8% purity, Merk KGaA, Darmstadt, Germany), and HCl (37% purity, Merk KGaA, Darmstadt, Germany) before tests as reported in a previous study [24,25].

Finally, two nanofluids were prepared with 500 mg·$L^{-1}$ AlNi1 and AlNi1Pd1, respectively. Each nanofluid also contains 0.1 % in mass fraction of solution of tween 80 (Panreac, Barcelona, Spain) in distilled water. The nanofluids were magnetically stirred at 300 rpm at 25 °C for six hours and then sonicated for four hours.

#### Porous Media

100 g of silica sand was compacted into a stainless-steel tube of 60 cm length, 2.54 cm inside diameter (ID), and 8 cm outside diameter (OD). A Stainless-steel tube withstands fluid flow pressure up to 10,000 psi and temperatures up to 300 °C. The porosity was determined through a saturation method at a fixed flow of 0.5 mL·$min^{-1}$ in a positive displacement pump [37]. The porous volume is 130.5 mL for both systems. The absolute permeability was measured by injecting brine in sand packed at 0.5 mL·$min^{-1}$. One pressure transductor (Rosemount, Emerson, MO, USA) was used to record the average system pressure. Darcy's law was used to estimate the permeability [37]. The high value of petrophysics properties tries to simulate the formation conditions where the steam is injected in the Colombia field [38].

*2.2. Methods*

Figure 1 shows the proposed workflow to develop and evaluate two doped nanoparticles in crude oil upgrading and recovery in displacement tests varying the steam quality (0.5 and 1.0). The protocol starts with the synthesis and characterization of nanoparticles and their subsequent evaluation at static conditions through batch adsorption isotherms and thermogravimetric analysis (TGA). The TGA tests were done at isothermal and non-isothermal conditions. Then, coreflooding experiments were executed by injecting steam at different qualities (0.5 and 1.0) into the porous medium. The processes were assisted by two nanofluids (AlNi1 and AlNi1Pd1). Finally, the effluents recovered in each stage were robustly analyzed by API gravity, Saturates, Aromatics, Resins, and Asphaltenes (SARA) distribution, and rheological behavior to determine the impact of steam quality and nanoparticles' chemical nature in crude oil upgrading.

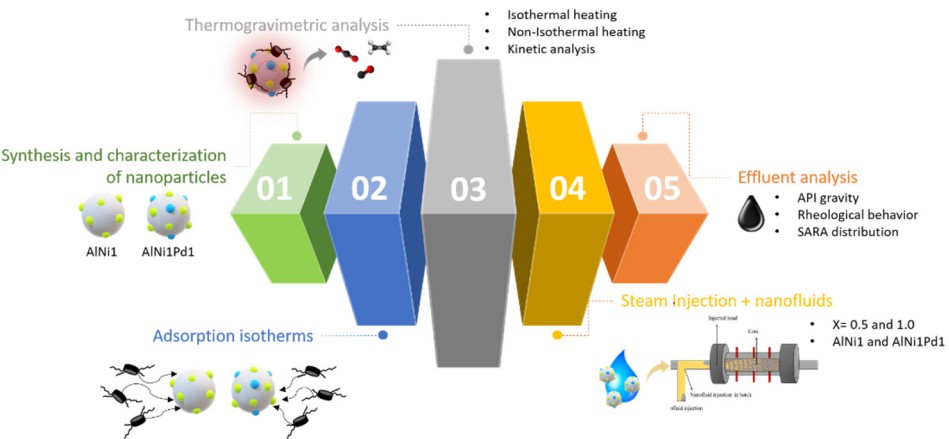

**Figure 1.** Workflow was conducted in this investigation.

2.2.1. Nanoparticles Characterization

The nanoparticles were characterized by mean particle size, surface area, as well as crystal size, and dispersion of Ni and Pd. The nanoparticles' mean size was measured with dynamic light scattering (DLS) using a Nanoplus-3 analyzer, (Micrometrics, Norcross, GA, USA), Scanning Electron Microscopy (SEM) using a Field Electron and Ion (FEI) microscope model Quanta 400 (SEM) (Eindhoven, The Netherlands). The microscopy technique was coupled to Energy Dispersive X-ray spectroscopy (EDX).

The surface area was measured by the Brunauer-Emmett-Teller surface method ($S_{BET}$) [39]. The $S_{BET}$ method measures the quantity of nitrogen adsorbed and desorbed for the nanoparticles at $-196\ ^\circ$C using an Autosorb-1 from Quantachrome after outgassing samples at 140 $^\circ$C under high vacuum for 12 h. The size and dispersion of Ni and Pd over the support ($Al_2O_3$) were evaluated using a Chembet 3000 (Quantachrome Instruments, Boynton Beach, FL, USA) by $H_2$ chemisorption. The protocol is described in detail in our previous work [26].

2.2.2. Adsorption Batch Experiments

The nanoparticles' adsorption capacity was evaluated by batch adsorption experiments at 25 $^\circ$C using asphaltene model solutions containing between 100 and 2000 mg·L$^{-1}$ in toluene. The colorimetry method was employed to obtain the adsorption amount. The protocol is described in detail in several works [22,23,26,40]. The mass balance showed in Equation (1) was used to determine the adsorbed amount of *n*-C$_7$ asphaltenes.

$$q = \left( \frac{C_O - C_E}{A} \right) M \tag{1}$$

where, $C_E$ is the asphaltenes equilibrium concentration in the supernatant (mg·L$^{-1}$), $C_O$ is the initial concentration of asphaltene in the solution (mg·L$^{-1}$), $A$ is the surface area of nanoparticles (m$^2$), and $M$ is the mass of adsorbent (mg). Adsorption isotherms were adjusted using the Solid-Liquid equilibrium model (SLE) [18,26].

### 2.2.3. Catalytic Decomposition of Asphaltenes

The nanoparticles' catalytic activity was evaluated in a thermogravimetric analyzer Q50 (TA Instruments, Inc., New Castle, DE, USA). The equipment was used to simulate the steam gasification of asphaltenes in the absence and presence of nanoparticles by injecting 100 mL·min$^{-1}$ of N$_2$ and 6.30 mL·min$^{-1}$ of H$_2$O$_{(g)}$ using a gas saturator filled with distilled water. The evaluation was done at non-isothermal conditions in the temperature range of 100–600 °C at a heating rate of 10 °C·min$^{-1}$ and isothermal heating using three temperatures (210 °C, 220 °C, and 230 °C). The asphaltene adsorbed amount for catalytic experiments was fixed at 0.2 mg·m$^{-2}$ ± 0.02 mg·m$^{-2}$. It is worth mentioning that during static experiments, the steam quality was not evaluated. The effective activation energy for thermogravimetric experiments was estimated using an isothermal solution model described in previous works [22,26].

### 2.2.4. Displacement Tests

The coreflooding tests were performed to evaluate the steam quality effect on oil upgrading and recovery assisted with bimetallic and monometallic nanoparticles. The test consists of six stages, including, (i) basic characterization, (ii) waterflooding, (iii) steam injection at 0.5 quality (X = 0.5), (iv) steam injection at 1.0 quality (X = 1.0), (v) batch injection of nanofluids, (vi) steam injection after nanofluid injection at 0.5 and 1.0 qualities. Batch injection introduces a determined amount of liquid (nanofluid) into the porous medium at a defined injection rate. This technique does not use other agents, such as gases, for dispersion of the nanofluid. Figure 2 shows a graphic representation of batch nanofluid injection.

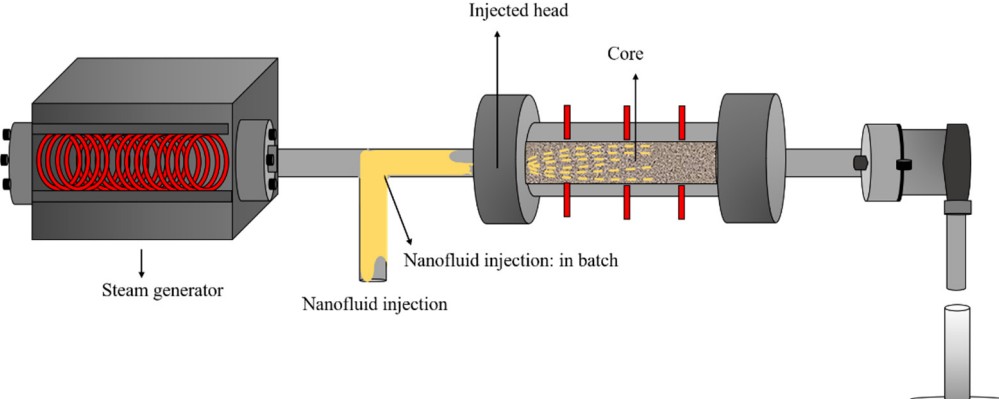

**Figure 2.** Graphical representation of nanofluid injection in batch into the porous medium.

Figure 3 shows the experimental assembly used for the displacement tests. It comprises two positive displacement pumps (DB Robinson Group, Edmonton, AB, Canada), oil, brine, water, and nanofluid containing cylinders, a tubular furnace (Thermo Scientific Waltham, MA, USA). The core holder (3.8–50 cm long) is equipped with six thermocouples (Termocuplas, S.A.S., Medellín, Colombia) and a hydraulic pump (Enercap, Actuant Corporation, WI, USA). Also, the system contains a pressure multiplier and a separator.

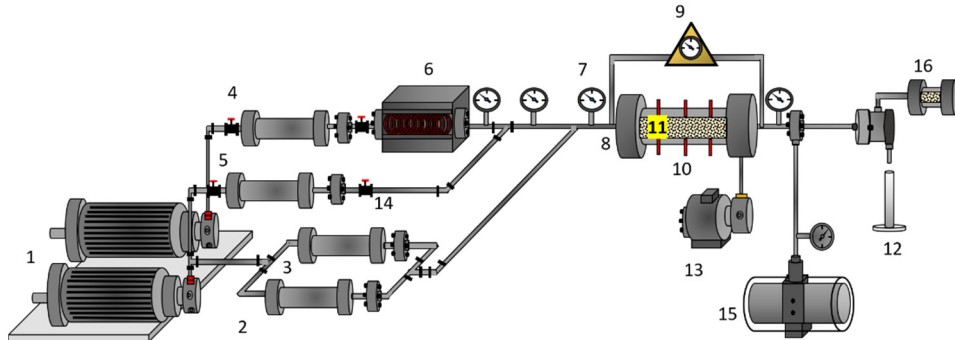

**Figure 3.** Steam generation and displacement system for experimental tests. Legend: (1) Positive displacement pumps, (2) oil-containing displacement cylinder, (3) brine-containing displacement cylinder, (4) water-containing cylinder, (5) nanofluid-containing cylinder, (6) tubular furnace, (7) manometers, (8) thermocouple, (9) pressure transducer, (10) slim tube, (11) sand packed bed, (12) sample output, and (13) hydraulic pump, (14) valves, (15) pressure multiplier and (16) gas collector. Reproduced with permission from Medina et al. [24], Energies, MDPI; Published by MDPI, 2019.

The basic characterization of the porous media was done at a back pressure of 1.03 MPa (150 psi), overburden pressure of and 5.51 MPa (800 psi), and temperature of 80 °C. The absolute and effective permeabilities ($K_{abs}$ and $k_o$) are estimated using Darcy's law [41]. For calculating the absolute permeability, 10 pore volumes (PV) of brine were injected into the porous media at a defined rate of 0.5 mL·min$^{-1}$. Afterward, the crude oil injection was done until the pressure stabilized. Then 20 PV of water were injected for determining the water effective permeability (Kw) at residual oil saturation (Sor) conditions [42]. The injection of AlNi1Pd1 and AlNi1 nanoparticles was evaluated in the porous medium 1 and 2, respectively. Table 1 presents the porous media properties.

**Table 1.** Petrophysics properties of porous media for the steam recovery processes in absence and presence of nanoparticles at X = 0.5 and X = 1.0 quality.

| System | Porous Medium 1 | Porous Medium 2 |
|---|---|---|
| Mineralogy | Silica 99% | Silica 99% |
| Length (cm) | 60 | 60 |
| Diameter (cm) | 2.5 | 2.5 |
| Porous volume (mL) | 131.5 | 132 |
| Porosity (%) | 38 | 38 |
| Absolute permeability | 9080 | 9100 |
| Nanoparticle injected (500 mg·L$^{-1}$) | AlNi1Pd1 | AlNi1 |

The values presented in Table 1 are representative of simulating steam injection conditions. Besides, both porous media characteristics indicate that the results are comparable. Then, the water flooding experiments were done by introducing the brine from the water cylinder through the coil line into the porous media until residual oil saturation ($S_{or}$) conditions are achieved. The water was injected at a flow ranging between 2.5 mL·min$^{-1}$ and 6.5 mL·min$^{-1}$. The temperature during this stage was the reservoir temperature (70 °C). The temperature during this stage was the reservoir temperature (70 °C). According to our previous studies, this temperature is sufficient to promote the mobility of the crude oil in the porous medium [43,44].

The incremental crude oil produced was estimated, and then steam was injected at X = 1.0 until there was no oil production. At this point, close to 15 and 16 PV of water equivalent to steam has been injected. Next, 0.5 PV of a hydrogen donor is placed in the porous media during a soaking time of 4 h. Subsequently, 0.5 PV of the nanofluid is injected in batch at 0.5 mL·min$^{-1}$ and left to act for 4 h with the reservoir fluids. For both nanofluid presence scenarios, steam is again injected at X = 0.5 between 3 mL·min$^{-1}$ and 5 mL·min$^{-1}$ until no

more oil is produced. Then, X was raised to 1 to verify the incremental oil production in the presence of nanoparticles. The dosage and soaking time of the nanofluids were selected based on previous works [21,24,25].

Finally, the steam is injected at 210 °C and 1.90 MPa (276 psi) and 1.44 MPa (210 psi) to ensure a steam quality of 0.5 and 1.0, respectively. The steam quality proposed in the experimental methodology was guaranteed using a model based on heat transfer equations, and mass and energy conservation balances were made, identifying steam change through the generation and injection system. Details are found in Section S1 of the Supplementary material information.

### 2.2.5. Effluent Analysis

The effluents obtained during the displacement test were characterized by API gravity, rheological behavior, SARA distribution, and residue content. The API gravity was measured using the ASTM D369 standard in an Anton Paar Stabinger SVM 3000 rotating cylinder viscometer (Madrid, Spain). The saturates, aromatics, resins, and asphaltene (SARA) content of the effluents was determined following the ASTM D6560 standard by combining the IP 469 method and micro-deasphalting with *n*-heptane using an TLC-FID/FPD Iatroscan MK6 (Iatron Labs Inc., Tokyo, Japan) [45,46]. Simulated distillation (SimDis) was done to determine the residue conversion (R%). The residue content (620 °C+) was estimated using high-temperature simulated distillation (HTSD) following the ASTM D-7169 [47] in a gas chromatograph equipped with a capillary (Agilent, Needle 0.25 mm on the column, 5 uL syringe 3/PK) provided by Agilent Technologies (Wilmington, DE, USA). Results are given as residue conversion (R%) as Equation (2) shows:

$$R\% = \left( \frac{R_{virgenEHO} - R_{treatment}}{R_{virgenEHO}} \right) \times 100 \qquad (2)$$

$R_{virgenEHO}$ and $R_{treatment}$, refer to the residue content before and after steam injection, respectively. Finally, the effluents are characterized by viscosity and rheology measurements using a Kinexus Pro+ rheometer (Malvern Instruments, Worcestershire, UK) using a gap of 0.3 mm serrated plate-plate geometry for a shear rate range between 0 s$^{-1}$ to 100 s$^{-1}$ at 25 °C. The degree of viscosity reduction (DVR) was determined according to Equation (3):

$$DVR\% = \left( \frac{\mu_{ref} - \mu_f}{\mu_{ref}} \right) \times 100 \qquad (3)$$

where, $\mu_{ref}$ and $\mu_f$ are the reference viscosity and viscosity after treatment, respectively. All experiments were performed for triplicate. Finally, rheology behavior was modeled with the Cross model [48]. Modeling details can be found in Section S1 of the Supplementary Materials.

## 3. Results

### 3.1. Nanoparticles Characterization

The morphology of the AlNi1Pd1 and AlNi1 samples was evaluated through Scanning Electron Microscopy, and the micrographs are shown in panels (a, b) of Figure 4. Non-morphological particles with different fusion degrees form the alumina network. Monometallic and bimetallic samples present similar structures with interconnected porous particles of different sizes. The AlNi1Pd1 and AlNi1 average size by SEM analysis was around 99 nm and 85 nm, respectively. This was corroborated by DLS where mean particles size of 80 nm and 96 nm were obtained. The size measurements of the active phases (Ni and Pd) further confirmed this. Figures 5 and 6 show the EDX results for AlNi1 and AlNi1Pd1, respectively. It was observed that the metal percentage of Ni was 1.9% in mass fraction in AlNi1, and 1.0% in mass fraction in AlNi1Pd1. The Pd percentage was 0.98% in mass fraction for AlNi1Pd1. The results were similar to the theoretical calculated for AlNi1 (2.0%

in mass fraction of Ni) and AlNi1Pd1 (1.0% in mass fraction of Ni and 1.0% in mass fraction of Pd). The rest of the samples is composed of alumina.

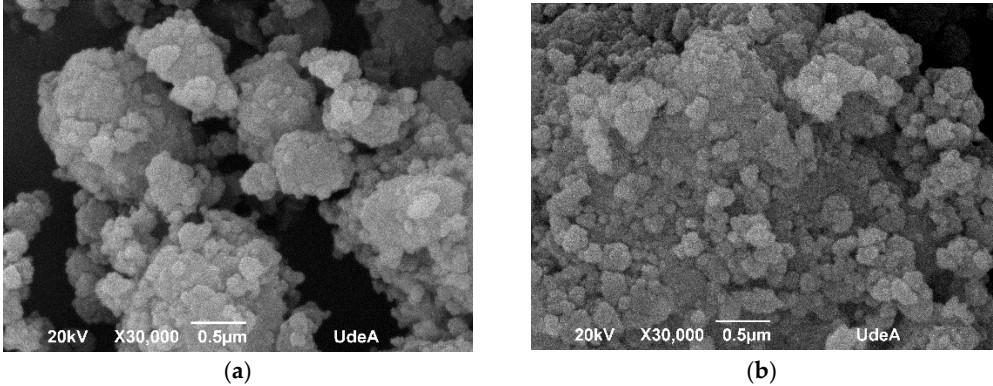

(**a**)　　　　　　　　　　　　　　　　　(**b**)

**Figure 4.** Scanning electron microscopy images of (**a**) AlNi1 and (**b**) AlNi1Pd1.

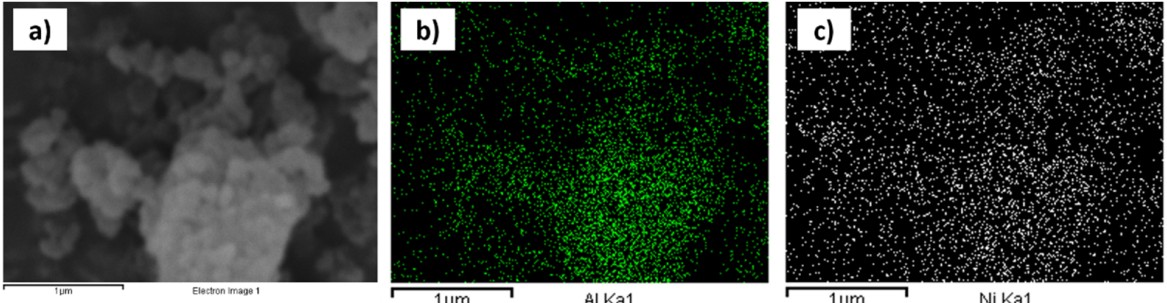

**Figure 5.** Energy-Dispersive X-ray images of AlNi1. (**a**) SEM image, (**b**) Al content, and (**c**) Ni content.

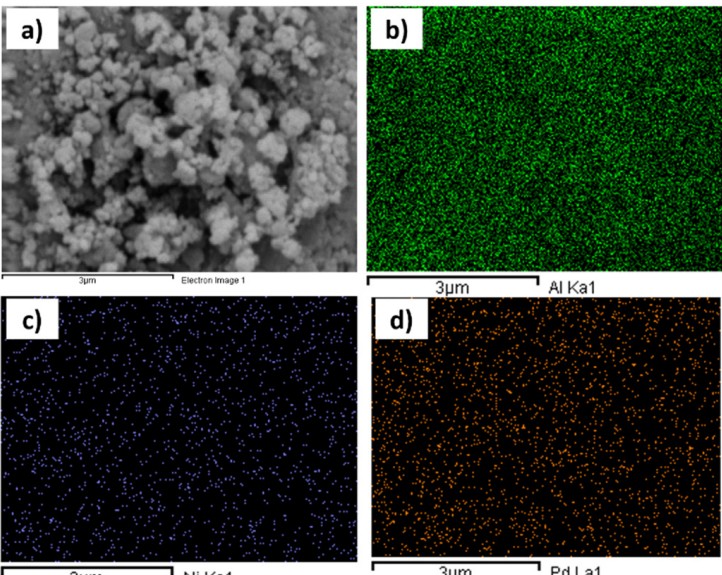

**Figure 6.** Energy-Dispersive X-ray images of AlNi1Pd1. (**a**) SEM image, (**b**) Al content, (**c**) Ni content, and (**d**) Pd content.

The surface area of the support was 225 m$^2 \cdot$g$^{-1}$. After the functionalization process, the surface area decreased to 223.4 m$^2 \cdot$g$^{-1}$ and 65.9 m$^2 \cdot$g$^{-1}$ for AlNi1 and AlNi1Pd1, respectively. The reduction of the surface area was due to the blocking pores for the impregnation of the precursor salts onto the surface of the nanoparticle. These results

agree with the reported elsewhere for porous systems doped with transition element oxides [18,23,28,40].

The size and dispersion of the crystals of Ni and Pd were obtained through $H_2$ pulses and are summarized in Table 2. The Ni dispersion was higher in the bimetallic system because the presence of Pd in the system avoids the sintering process of Ni crystals [26,28]. Also, a higher distribution means a smaller size of the active phase. Results agree with the reported by Medina et al. [26], which explains that Ni and Pd diffusion over the support surface is complex because of their high Tamman's temperature [49].

**Table 2.** Properties of nanoparticles AlNi1 and AlNi1Pd1 nanoparticles.

| Type of Nanoparticle | Pd (%) EDX | Ni (%) EDX | Dispersion (%) | | Average Crystal Size (nm) | |
|---|---|---|---|---|---|---|
| | | | Ni | Pd | Ni | Pd |
| AlNi1 | - | 1.9 | 4.48 | - | 4.50 | - |
| AlNi1Pd1 | 0.98 | 1.0 | 5.40 | 9.90 | 2.20 | 4.10 |

### 3.2. Adsorption Experiments

Figure S1 of the supplementary material shows the experimental adsorption isotherms of $n$-$C_7$ asphaltenes onto monometallic nanoparticles (AlNi1) and bimetallic nanoparticles (AlNi1Pd1) and the fitted SLE model. According to the International Union of Pure and Applied Chemistry (IUPAC), the adsorption isotherms obtained behaves like the Ib type. Between the systems, the adsorption amount at any equilibrium asphaltene concentration increases in the order AlNi1 < AlNi1Pd1. The presence of the bielemental oxides on Al support increases the adsorption capacity for asphaltene uptake. Table S2 of the supplementary material contains the estimated SLE parameters for each nanoparticle. It was obtained that Henry's law constant (H) follows the increasing order AlNi1Pd1 < AlNi1. As higher the H values, the lower the adsorbate-adsorbent affinity. In this sense, the presence of bimetallic nanoparticles increases the affinity for asphaltenes to a greater degree than monometallic systems, due to the fact of different active sites that lead to a higher selectivity for asphaltenes and higher intermolecular forces [50]. The results agree with those reported in previous work for asphaltene adsorption onto functionalized alumina nanoparticles [28].

### 3.3. Thermogravimetric Analysis

#### 3.3.1. Non-Isothermal Thermogravimetric Experiments

The nanoparticles' catalytic activity was evaluated through thermogravimetric analysis to determine the lowest peak temperature in which asphaltene decomposition occurs. Figure 7 shows the results obtained. The non-catalyzed system shows that the virgin asphaltene gasification begins at 350 °C and ends at 550 °C with a maximum loss at 472 °C approximately. The asphaltenes are high weight molecules that do not have a mass loss significantly at low temperature. The asphaltenes can have a gaseous conversion, due to the partial pyrolysis and evaporation of light molecular weight hydrocarbons [51]. It is reported that asphaltene molecules isolated from different crude oils decompose at high temperatures between 400 and 500 °C [19,23,52].

The support was not evaluated as it demonstrated its low catalytic activity towards asphaltenes decomposition at low temperatures [29]. The catalyzed systems show a displacement of the main decomposition peak to lower temperatures of 250 °C and 230 °C for AlNi1 and AlNi1Pd1 nanoparticles, respectively. The catalytic activity is correlated with the presence of monometallic and bimetallic phases on the $Al_2O_3$ surface. The presence of Ni and Pd increases the active sites for $H_2O_{(g)}$ anchorage, and therefore, higher interactions with asphaltenes [22,25].

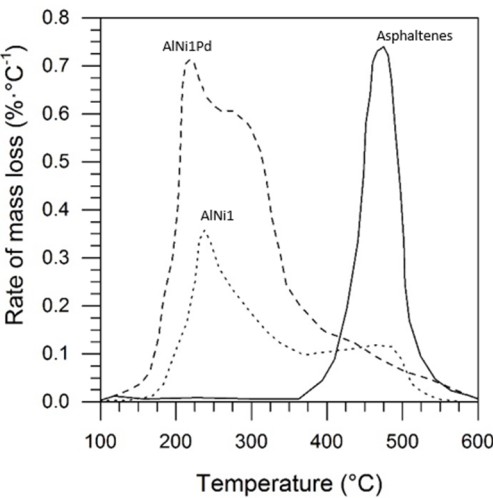

**Figure 7.** Rate of mass loss for non-catalyzed and catalyzed *n*-C$_7$ asphaltene gasification by AlNi1 and AlNi1Pd1 nanoparticles.

The first decomposition peak would correspond to the opening of the availability of polycyclic aromatic hydrocarbons of the asphaltenes. The second peak would correspond to the decomposition of heavier molecules [36,53].

It is evidenced that the materials' catalytic activity agrees with their affinity for asphaltenes and the reduction of the self-association degree on the catalyst surface. It is concluded that individually adsorbed molecules have lower energy requirements for decomposition than adsorbed aggregates [28].

### 3.3.2. Isothermal Thermogravimetric Experiments and Kinetic Analysis

Considering the non-isothermal results and the average steam injection temperature in Colombian fields, isothermal experiments were performed at 200 °C, 210 °C, and 220 °C. The virgin asphaltenes were subjected to 360 °C, 370 °C, and 380 °C, to ensure their conversion [26]. Figure 8 shows the results obtained. The conversion degree was higher for the AlNi1Pd1 system, followed by AlNi1 and the non-catalyzed system. Although virgin asphaltenes were heated at higher temperatures, their conversion was always lower than in the presence of nanoparticles. It reflects the capacity of Al-based systems to improve the conversion of asphaltene under a gasifying atmosphere. The AlNi1Pd1 and AlNi1 achieve 90% and 87% of asphaltene conversion at 150 min and 180 min, respectively, at 210 °C [29]. It is observed that all samples increase the conversion degree when the temperature rises. The presence of metal oxides also increases the degree of conversion. The activation energy and reaction kinetics were evaluated to compare the effect of metals in nanoparticle catalytic activity, and the results are summarized in Table 3.

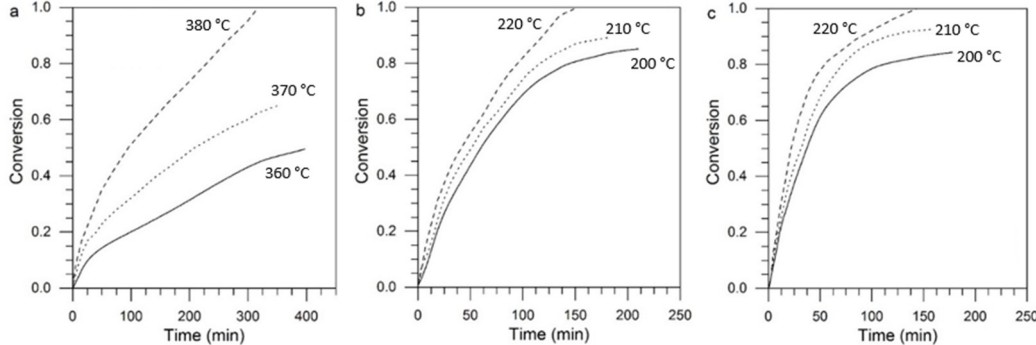

**Figure 8.** Isothermal conversion ($\alpha$) for steam gasification of (**a**) *n*-C$_7$ asphaltenes in the absence (360 °C, 370 °C, and 380 °C) and presence (200 °C, 210 °C, and 220 °C) of (**b**) AlNi1 and (**c**) AlNi1Pd1 samples.

**Table 3.** The estimated effective activation energy ($E_\alpha$) and kinetic rate for isothermal catalytic steam gasification of asphaltenes in the absence and presence of AlNi1 and AlNi1Pd1 samples.

| Material | Temperature (°C) | $E_\alpha$ (kJ) | $\frac{d\alpha}{dt}$ at 50% Conversion |
|---|---|---|---|
| Virgin Asphaltenes | 360 | | 0.012 |
| | 370 | 211.41 | 0.018 |
| | 380 | | 0.032 |
| AlNi1 | 200 | | 0.039 |
| | 210 | 30.39 | 0.050 |
| | 220 | | 0.078 |
| AlNi1Pd1 | 200 | | 0.083 |
| | 210 | 64.65 | 0.090 |
| | 220 | | 0.127 |

The activation energy is higher in AlNi1Pd1 than AlNi1, and the value depends on differences in reaction mechanisms; for this reason, the kinetic reaction rate is necessary for making the comparison. The asphaltenes conversion time is reduced by 44.0% in the presence of AlNi1Pd1 compared to AlNi1.

The AlNi1Pd1 presents a higher value for activation energy than AlNi1, due to the higher number of molecules to decompose. Even though AlNi1Pd1 gives that result, the reaction still has the fastest reaction rate ($\frac{d\alpha}{dt}$). This means that asphaltenes are cracked faster by AlNi1Pd1 than AlNi1, which is related to the low entropy produced, due to the low molecular disorder of the adsorbed compounds (K) in AlN1Pd1. This result agrees with the adsorption affinity and catalytic activity through the thermogravimetric analysis [54].

*3.4. Displacement Test*

The steam injection at different qualities was simulated through displacement tests. From the model described in Section S1 of the Supplementary Material, the steam generation conditions were adjusted, considering maximum steam quality losses, due to heat transfer of 3%. In this way, steam injection at X = 0.5 was done at 210 °C, and 1.90 MPa (276 psi) and steam injection at X = 1.0 was achieved at 210 °C and 1.44 MPa (210 psi).

3.4.1. Waterflooding and Steam Injection without Nanoparticles at Different Qualities

As it was described in Section 2.2.4, the first stage consisted of waterflooding. During this step, around 55% of the original oil was recovered in both porous media. Then, during the steam injection at X = 0.5, the oil recovery percentage increased by 3.0% in all cases from waterflooding stage. Then, the steam was injected at X = 1.0, obtaining an incremental of 2% in both porous media concerning the X = 0.5 stage and 5% regarding waterflooding stage. This result is consistent with experimental and simulation works for steam injection at different qualities [10,54–56]. When constant pressure is maintained in the reservoir, as higher steam quality, higher the heat transfer yield from steam to the reservoir fluids; in this sense, crude oil viscosity could be reduced to a higher degree at X = 1.0 than 0.5. It is reported that high steam quality leads to an optimal approach for better recoveries, due to a reduction of viscosity and physical displacement of in-situ oil [11,12]. Other problems like viscous fingering and steam override can be overcome [10]. Based on the heat lost in both scenarios when porous media is saturated with oil, it is obvious the effect of steam quality in increasing the oil recovery. The loss of heat inside the porous media was calculated through the heat transfer model (Equation (S1), Equation (S6), Equation (S7), and Equation (S8)), predicting the properties of the steam with the liquid and saturated steam states. For steam injection at X = 0.5, the steam quality was reduced to 0.17 in the first 15 cm of length in the saturated porous media. After 30 cm, the steam is found in a saturated liquid state. In the other scenario, when X = 1.0, 10% of energy was lost during the first 15 cm, obtaining a steam quality of 0.9. These results indicate that in the case X = 0.5, the heat losses are

more significant in the first sections of the porous medium, hindering an effective transfer of heat to the fluids.

### 3.4.2. Steam Injection with Nanoparticles at Different Qualities

The nanofluids containing AlNi1 and AlNi1Pd1 nanoparticles were injected together with a 0.5 PV of a hydrogen donor. When steam was injected at X = 0.5, the oil recovery increased 20.0% and 13.0% for AlNi1Pd1 and AlNi1, respectively. The injection of the hydrogen donor was evaluated in a similar scenario in the absence of nanoparticles. The results show an increase in oil recovery lower than 0.3% at X = 0.5 and 1.0. The use of the hydrogen donor in thermal recovery processes minimizes the polymerization and addition reactions of cracked heavy crude oil molecules [57]. In this sense, the role of the hydrogen donor is enhanced by the presence of nanoparticles that are constantly cracking the crude oil molecules, allowing the interaction of the hydrogen donor and free radicals.

When the crude oil production stops, the steam quality changes to X = 1, hoping to increase the production. However, it seems that both systems evacuated the mobile oil with a steam injection of X = 50%. The final state of saturation after recovery with treatment train and AlNi1 and AlNi1Pd1 nanoparticles corresponds to the order of 26 and 20%, respectively. These results show that the nanoparticles increase the oil recovery even at low steam qualities. Nanoparticles improve the conductivity of the porous media, the injected and reservoir fluids, and hence, the heat transfer using a low steam quality [24]. The metal oxides (Ni and Pd) also improve the heat absorption capacity, obtaining a better yield for the AlNi1Pd1 than AlNi1 [58,59]. Metal oxides can absorb heat for a long time when exposed to a surface. In the case of AlNi1Pd1, there are a more significant number of active sites that contribute this property to the system, for which the heat transfer becomes more effective and generates higher yields than in AlNi1 [24].

Figure 9 shows oil recovery curves for the described scenarios. The increase in oil production in AlNi1 and AlNi1Pd1 is also possibly due to a change in the rock wettability, making the medium more water wettable, increasing the oil mobility [42]. The nanoparticles also reduce the asphaltene decomposition temperature, promoting reduced viscosity and improving oil quality [42,51,60–62].

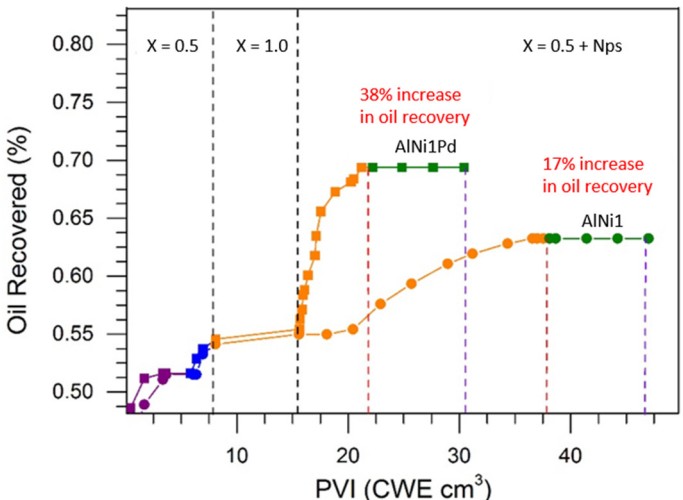

**Figure 9.** Oil recovery curve profile for steam injection without and with AlNi1 and AlNi1Pd1 nanoparticles at different steam qualities (X = 0.5 and X = 1.0).

Between both catalyzed scenarios, there is a big difference in oil recovery and water consumption. The system catalyzed by AlNi1Pd1 increases the oil recovery by 28.0% and reduces the water consumption by 25%. This result can be attributed to the fact that palladium can adsorb hydrogen on its surface up to 60% by weight [63], increasing the consumption rate of asphaltenes and stabilizing free radicals.

Another important fact is the reduction in water consumption. The porous volumes of water equivalent (PVWE) injected up to X = 1.0 in the absence of nanofluids were similar in both cases. Then, the injection of AlNi1 requires around 40 PVWE to achieve an increase in oil recovery of 17.0%, whereas the AlNi1Pd1 obtain a similar result by injecting around 17 PVWE. Finally, with 25 PVWE, a rise of 38% in oil recovery was obtained, demonstrating the potential of the AlNi1Pd1 nanocatalyst to assist this process.

### 3.5. Effluent Characterization

The recovered effluents after steam injection at different qualities in the absence and presence of AlNi1 and AlNi1Pd1 were characterized by API gravity, saturates/aromatics/resins/asphaltenes content, residue content, and rheological behavior to determine the effect of steam quality and nanofluid on crude oil quality.

#### 3.5.1. API Gravity Changes

The change in API gravity is shown in Figure 10. The virgin crude oil presents an API gravity of 6.9° at 25 °C, whereas after the steam is injected at X = 0.5 and 1.0, the API increases to 7.4° and 7.9°, respectively. The rise in steam quality increases the heat transfer to the reservoir fluids and the rock, achieving a subtle change in this property resulting from a possible rearrangement of the fluid molecules. After the catalyzed steam injection (X = 0.5) by AlNi1 and AlNi1Pd1 samples, API gravity was increased to 11.0° and 13.3°, respectively. The API gravity change is caused by converting adsorbed asphaltene molecules on the surface of the nanoparticle into lighter molecules. AlNi1Pd1 nanoparticles improve the oil quality more than AlNi1 nanoparticles, demonstrating the synergistic effect between nickel and palladium in the catalysis processes compared to nickel at the same temperature conditions [21,64,65]. The effluents obtained at X = 1.0 in the presence of AlNi1 and AlNi1Pd1 were similar to those obtained at X = 0.5.

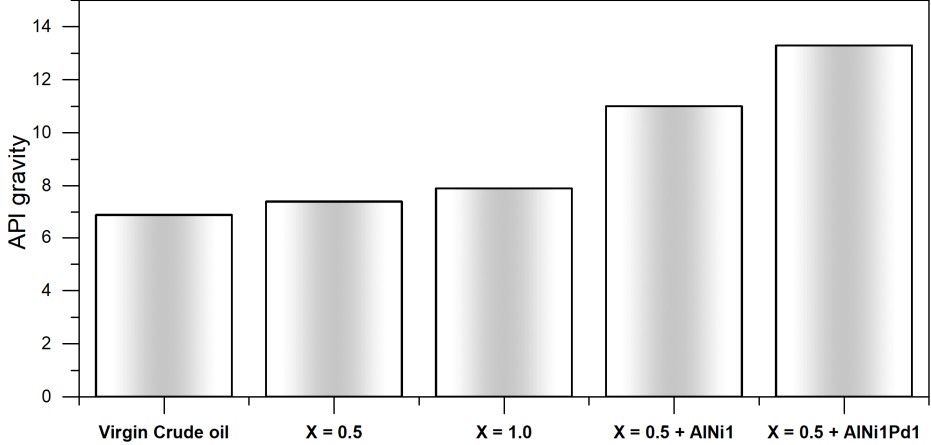

**Figure 10.** API gravity for untreated extra heavy oil and crude oil recovered after the steam injection at X = 0.5, X = 1.0, and X = 0.5 + AlNi1 and AlNi1Pd1.

#### 3.5.2. SARA Composition Changes

The distribution of SARA components was also changed in each stage analyzed and results are shown in Figure 11. The content of saturates was increased 2.0%, 4.0%, 4.5%, and 23.0% during the non-catalyzed steam injection at X = 0.5, X = 1.0, catalyzed steam injection at X = 0.5 in presence of AlNi1 and AlNi1Pd1, respectively. In the opposite way, the content of asphaltenes was reduced in the order X = 0.5 < X = 1.0 < X = 0.5 + AlNi1 < X = 0.5 + AlNi1Pd1. Like API gravity, the injection of steam at X = 1.0 in presence of AlNi1 and AlNi1Pd1 do not change the crude oil quality regarding the X = 0.5 + nanoparticles.

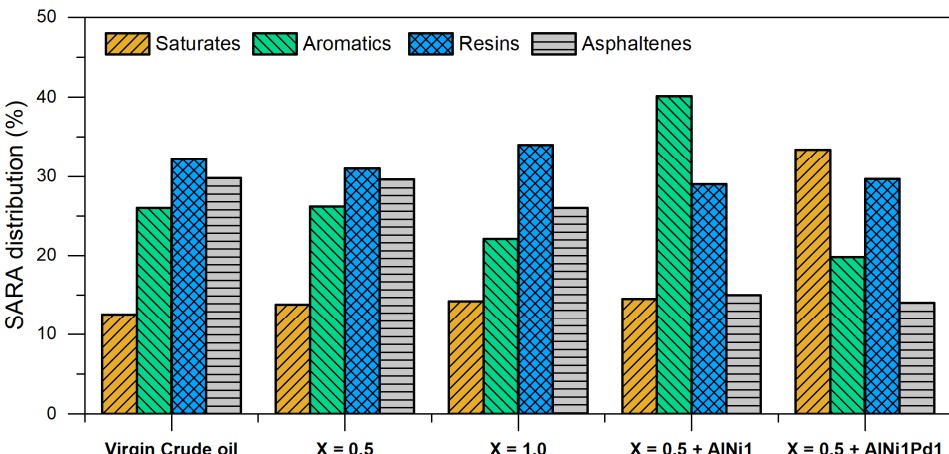

**Figure 11.** SARA content for untreated extra heavy oil and crude oil recovered after the steam injection at X = 0.5, X = 1.0, and X = 0.5 + AlNi1 and AlNi1Pd1.

The catalytic effect of the functionalized nanoparticles is noted. Both phases' synergistic effect is evidenced in the high decrease in asphaltene content in the system AlN1Pd1. Two main factors can explain this result: (i) The combined selectivities and reactivities of both metals toward the asphaltene molecules; and (ii) the metal sintering avoid on the alumina surface, which leads to increasing the number of active sites available for the reaction [28]. Finally, the species –O and –OH resulting from the dissociative adsorption of steam by the alumina lower valence state can be transferred to nickel and palladium and react with carbonaceous species at the surface [24].

### 3.5.3. Simulated Distillation

Throughout the results obtained in the simulated distillation, the nanoparticles' effect on the residue conversion at higher temperatures regarding the crude oil recovered after the injection of steam at different qualities.

The residue conversion follows the increasing order: Crude oil at X = 0.5 < crude oil at X = 1.0 < crude oil at X = 0.5 + AlNi1 < crude oil at X = 1.0 + AlNi1Pd1. In each system the obtained %R was 3.3.%, 3.7%, 31.0%, and 48.7%, respectively. The increase in steam quality at the experimental conditions evaluated generates a minimal change in the formation of lower molecular weight hydrocarbons. The AlNi1Pd1 sample inhibits the addition reactions to a higher degree than AlNi1 increasing the distilled fraction at lower temperatures. Based on these results, the crude oil increment obtained when X was raised from 0.5 to 1.0 could reduce crude oil viscosity, since the crude oil quality was not improved considerably.

### 3.5.4. Rheological Behavior Analysis

The rheological behavior of the virgin oil and the recovered with the non-catalyzed and catalyzed steam injection are studied, and the results are shown in Figure 12. It is noted that the viscosity values drop exponentially with an increase in shear rate, indicating a pseudoplastic behavior of the virgin extra heavy oil before and after steam injection. When heat is introduced in the porous media, the viscosity drops, due to the suppression of viscous force by the system's kinetic energy [66]. That explains the higher reduction in viscosity at X = 1.0 than at X = 0.5. After the injection of AlNi1 and AlNi1Pd1 nanoparticles, it is appreciated a considerable decrease in viscosity associated with the viscoelastic network disruption.

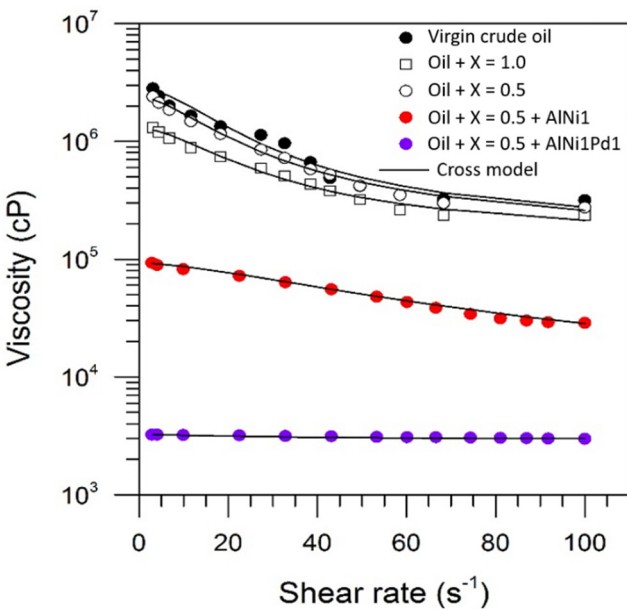

**Figure 12.** Recovered fluids rheological behavior at 25 °C. Virgin extra-heavy oil and oil recovered with steam injection at X = 0.5 and 1.0 and catalytic steam injection at X = 0.5 assisted by AlNi1 and AlNi1Pd1 samples.

Based on the behavior of viscosity as a function of the shear rate, it is observed that the virgin crude oil and the effluents obtained after the steam injection at 0.5 and 1.0 quality have a non-Newtonian behavior, where the viscosity decreases by increasing the shear rate. However, the AlNi1Pd1 and AlNi1 samples could modify the crude oil's rheological behavior from a pseudoplastic fluid to a Newtonian fluid, reducing the change in viscosity with increased shear rate. This result is due to the internal structure rupture of the crude oil and the posterior redistribution of the $n$-C$_7$ aggregate asphaltenes. The oil viscosity reduction was more pronounced in the catalyzed system by AlNi1Pd1. This results from the higher affinity for heavier fractions for Ni, Pd, and Al sited, reducing the asphaltene-asphaltene interaction by increasing nanoparticle-asphaltene interactions [67]. The Cross model fitted the rheological profiles, and the estimated parameters are shown in Table 4. For all profiles, the *RMS*% was lower than 2%, which indicates a good adjustment of the model to the rheological behavior. The viscosity parameters at extremes (i.e., at zero and infinity) decrease with the increase in steam quality and further with the addition of AlNi1 and AlNi1Pd1.

**Table 4.** Effect of steam injection at different qualities X = 0.5 and X 1.0 and catalytic steam injection at X = 0.5 in the presence of AlNi1 and AlNi1Pd1 in viscosity reduction degree (DVR) of an extra-heavy oil at 25 °C—the shear rate of 10 s$^{-1}$ and rheological parameters of the simulated model of Cross.

| Parameters Cross Model | Virgin Crude Oil | Oil + X = 50% | Oil + X = 100% | Oil + X = 50% + AlNi1 | Oil + X = 50% + AlNi1Pd1 |
|---|---|---|---|---|---|
| DRV % | - | 8.9 | 32.5 | 92.4 | 99.0 |
| $\mu_{inf} \times 10^3$ (cp) | 184 | 162 | 153 | 3.5 | 2.7 |
| $\mu_0 \times 10^3$ (cp) | 2811 | 2390 | 1302 | 93 | 3 |
| %RMS | 0.143 | 0.062 | 0.071 | 0.051 | 0.041 |

Finally, Table 4 shows the reduction viscosity percentage for a shear rate value of 10 s$^{-1}$ comparing with the untreated extra heavy oil. As can be seen, as the steam quality increases from X = 0.5 to X = 1.0, a difference in viscosity reduction of 23.6% can be achieved. For the case of nanofluids' presence, there is a reduction of more than 90% for both. However, in

the presence of the AlNi1Pd1 nanoparticles, an improvement in viscosity of 99% can be achieved with a Newtonian behavior.

## 4. Conclusions

- The steam injection at 0.5 and 1.0 quality increased the oil recovery by 3.0% and 7.0%, respectively, regarding the base curve (waterflooding). After nanofluid injection, the steam at 0.5 quality achieved an increase in oil recovery of 20.0% and 13.0% for AlNi1 and AlNi1Pd1 nanoparticles, respectively. When steam was injected at 1.0 quality for both nanoparticles evaluated, no incremental oil was produced. The adsorption capacity was determined through batch adsorption experiments, obtaining higher asphaltene adsorption for the AlNi1Pd1 sample.
- The catalytic experiments show that AlNi1 and AlNi1Pd1 samples achieve 100% conversion of adsorbed asphaltenes at 220 °C during 140 and 175 min, respectively.
- All functionalized nanoparticles tested in this work exhibit low energy activation compared with the virgin asphaltenes and a high gasification rate. The reaction mechanism depends on the metal's nature in the nanoparticles' surface and the interaction between different metals on the same surface.
- The change in steam quality did not affect the crude oil upgrading (API and SARA content). However, AlNi1 and AlN1Pd1-based nanofluids increased de API gravity from 6.9° to 11°, and 13.3° and the asphaltene content was reduced near to 30% and 50%, respectively, and the highest viscosity reduction percentage was 99% in for AlNi1Pd1.

**Supplementary Materials:** The following are available online at https://www.mdpi.com/article/10.3390/pr9061009/s1, Figure S1. Asphaltene adsorption isotherm over monometallic nanoparticles AlNi1 (nM), and bimetallic nanoparticles AlNi1Pd1 (nB) at 25 °C. The solid lines are the SLE model, and the symbols are the experimental data. Table S1. Estimated parameters of the SLE model for the asphaltene adsorption isotherms over AlNi1, and AlNi1Pd1 nanoparticles.

**Author Contributions:** Conceptualization, L.C., S.H.L., F.B.C. and C.A.F.; methodology, L.C. and O.E.M.; software and validation, S.C.; formal analysis, O.E.M.; investigation and data curation L.C. and O.E.M.; writing—original draft preparation, L.C. and O.E.M.; writing—review and editing, All authors. All authors have read and agreed to the published version of the manuscript.

**Funding:** This work was funded by COLCIENCIAS and Agencia Nacional de Hidrocarburos (ANH) through the Agreement 272-2017.

**Institutional Review Board Statement:** Not applicable.

**Informed Consent Statement:** Not applicable.

**Data Availability Statement:** Not applicable.

**Acknowledgments:** The authors acknowledge Petroraza S.A. and the Universidad Nacional de Colombia for logistical and financial support. Also, the authors thank Daniela Arias Madrid for her contribution in the development of this work.

**Conflicts of Interest:** The authors declare no conflict of interest.

## Abbreviations

| | |
|---|---|
| AlNi1 | Alumina nanoparticles doped with 1.0% in mass fraction of Ni. |
| AlNi1Pd1 | Alumina nanoparticles doped with 1.0% in mass fraction of Ni and Pd. |
| TEOR | Thermal Enhanced Oil Recovery |
| HO | Heavy oil |

| | |
|---|---|
| EHO | Extra-heavy oil |
| $S_{BET}$ (m$^2$) | Surface area estimates by the Brunauer-Emmett-Teller surface method |
| EDX | Energy Dispersive X-ray spectroscopy |
| SEM | Scanning Electron Microscopy |
| DLS | Dynamic Light Scattering |
| $K_{abs}$ | Absolute permeability |
| $k_o$ | Oil effective permeability |
| DVR | Degree of viscosity reduction |
| R% | Residue content |
| $\mu_{ref}$ (cP) | Reference viscosity |
| $\mu_f$ (cP) | Viscosity after treatment |
| $\mu_{inf}$ (cP) | Viscosity at the infinite shear rate |
| $\mu_0$ (cP) | Viscosity at zero shear rate |
| H (mg g$^{-1}$) | Henry's law constant |
| K (g g$^{-1}$) | Adsorbate association degree |
| $q_m$ (g g$^{-1}$) | Maximum adsorbed amount of adsorbate |
| $C_E$ (mg L$^{-1}$) | Equilibrium concentration |
| $C_O$ (mg L$^{-1}$) | Initial concentration of asphaltene in the solution |
| $T$ ($^\circ$C) | Temperature |
| $E_\alpha$ (kJ mol$^{-1}$) | Activation energy |
| X | Steam quality |
| $\frac{d_\alpha}{d_t}$ | Reaction rate |

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
