# Peer review of "Effect of Steam Quality on Extra-Heavy Crude Oil Upgrading and Oil Recovery Assisted with PdO and NiO-Functionalized Al2O3 Nanoparticles"

_processes, doi:10.3390/pr9061009_

Round 1

Reviewer 1 Report

The following issues should be modified.
1- Abstract section should be rewritten more scientifically

2- The following references are recommended to cite and discuss

  • Experimental investigation and mathematical modeling of gas diffusivity by carbon dioxide and methane kinetic adsorption
  • Thermodynamic effects of cycling carbon dioxide injectivity in shale reservoir
  • Significance of Aquathermolysis Reaction on Heavy Oil Recovery during the Steam-Assisted Gravity Drainage Process
  • New insight into CO2 huff-n-puff process for extraheavy oil recovery via viscosity reducer agents: An experimental study.
  • Parametric study of polymer-nanoparticles-assisted injectivity performance for axisymmetric two-phase flow in EOR processes

3- Literature review is really poor and should be compared to the novelty of your work one by one. For example, say ...et al... did this and we present this. why is different and what are the novelties. It is not acceptable to say just the novel points without any comparison.

4- The conclusion section should be improved

5- The nomenclature section is missing and should be added

6- Please add oil composition in the materials section

Author Response

Medellín, May 20, 2021

Ms. Susan Ji

MDPI

RE: Response to reviewer’s comments regarding manuscript ID: processes-1233476

Dear Ms. Susan Ji

We would like to thank you for securing a prompt review of our manuscript titled; “Effect of Steam Quality on Extra-Heavy Crude Oil Upgrading and Oil Recovery Assisted with PdO and NiO-Functionalized Al2O3 Nanoparticles.” We have answered all the comments raised by the reviewer and have thoroughly revised the manuscript accordingly. We found the comments helpful and believe that our revised manuscript represents a significant improvement over our initial submission.

The detailed response (in blue) to the reviewer’s comments, suggestions, and questions (in black), and the revised manuscript are attached. As suggested, any track changes, highlights, or font colors in our revised manuscript have been removed and we believe now that our manuscript is publishable in Processes Journal. 

Please do not hesitate to contact us if you have any further questions.

Sincerely yours,

The authors

Responses to the reviewer´s suggestions for the manuscript: Effect of Steam Quality on Extra-Heavy Crude Oil Upgrading and Oil Recovery Assisted with PdO and NiO-Functionalized Al2O3 Nanoparticles.

Reviewer 1.

Comment: The following issues should be modified. Abstract section should be rewritten more scientifically.

Response: We thank the reviewer for his/her effort made for securing a prompt review of our manuscript and for his/her suggestion. Based on the comment, the abstract was modified as follows:

“This work focuses on evaluating the effect of the steam quality on the upgrading and recovery of extra-heavy crude oil in the presence and absence of two nanofluids. The nanofluids AlNi1 and AlNi1Pd1 consist of 500 mg·L-1 of alumina doped with 1.0% in mass fraction of Ni (AlNi1) and alumina doped with 1.0% in mass fraction of Ni and Pd (AlNi1Pd1), respectively, and 1000 mg·L-1 of tween 80 surfactant. Displacement tests were done in different stages, including i) basic characterization, ii) waterflooding, iii) steam injection at 0.5 quality, iv) steam injection at 1.0 quality, v) batch injection of nanofluids, vi) steam injection after nanofluid injection at 0.5 and 1.0 qualities. The steam injection was realized at 210 °C, the reservoir temperature was fixed at 80 °C, and pore and overburden pressure at 1.03 MPa (150 psi) and 5.51 MPa (800 psi), respectively. After the steam injection at 0.5 and 1.0 quality, oil recovery was increased 3.0% and 7.0%, respectively, regarding the waterflooding stage, and no significant upgrade in crude oil was observed. Then, during the steam injection with nanoparticles, the AlNi1 and AlNi1Pd1 increased the oil recovery by 20.0% and 13.0% at 0.5 steam quality. Meanwhile, when steam was injected at 1.0 quality for both nanoparticles evaluated, no incremental oil was produced. The crude oil was highly upgraded for AlNi1Pd1 system, reducing oil viscosity 99%, increasing API ° from 6.9 ° to 13.3°, and reducing asphaltene content 50% at 0.5 quality. It is expected that this work will eventually help to understand the appropriate conditions in which nanoparticles should be injected in a steam injection process to improve its efficiency in terms of oil recovery and crude oil quality.

Comment: The following references are recommended to cite and discuss

  • Experimental investigation and mathematical modeling of gas diffusivity by carbon dioxide and methane kinetic adsorption
  • Thermodynamic effects of cycling carbon dioxide injectivity in shale reservoir
  • Significance of Aquathermolysis Reaction on Heavy Oil Recovery during the Steam-Assisted Gravity Drainage Process
  • New insight into CO2 huff-n-puff process for extraheavy oil recovery via viscosity reducer agents: An experimental study.
  • Parametric study of polymer-nanoparticles-assisted injectivity performance for axisymmetric two-phase flow in EOR processes

Response: We thank the reviewer for his/her suggestion. The references were added in the introduction section as shown below:

The HO and EHO reservoirs are targeted by non-thermal and thermal enhanced oil recovery technologies. The non-thermal methods include miscible [1-3] and non-miscible gas injection [4-6], while thermal methods particularly use air or steam flooding [7-9].”

“1.       Davarpanah, A.; Mirshekari, B. Experimental investigation and mathematical modeling of gas diffusivity by carbon dioxide and methane kinetic adsorption. Industrial & Engineering Chemistry Research 2019, 58, 12392-12400.

  1. Hu, X.; Xie, J.; Cai, W.; Wang, R.; Davarpanah, A. Thermodynamic effects of cycling carbon dioxide injectivity in shale reservoirs. Journal of Petroleum Science and Engineering 2020, 195, 107717.
  2. Li, S.; Lu, C.; Wu, M.; Hu, Z.; Li, Z.; Wang, Z. New insight into CO2 huff-n-puff process for extraheavy oil recovery via viscosity reducer agents: An experimental study. Journal of CO2 Utilization 2020, 42, 101312.
  3. Medina, O.E.; Olmos, C.; Lopera, S.H.; Cortés, F.B.; Franco, C.A. Nanotechnology Applied to Thermal Enhanced Oil Recovery Processes: A Review. Energies 2019, 12, 4671.
  4. Galeano-Caro, D.; Villegas, J.P.; Sánchez, J.H.; Cortés, F.B.; Lopera, S.H.; Franco, C.A. Injection of Nanofluids with Fluorosurfactant-Modified Nanoparticles Dispersed in a Flue Gas Stream at Very Low Concentration for Enhanced Oil Recovery (EOR) in Tight Gas–Condensate Reservoirs. Energy & Fuels 2020, 34, 12517-12526.
  5. Villegas, J.; Moncayo-Riascos, I.; Galeano-Caro, D.; Riazi, M.; Franco, C.A.; Cortés, F.B. Functionalization of γ-Alumina and Magnesia Nanoparticles with a Fluorocarbon Surfactant to Promote Ultra Gas-Wet Surfaces: Experimental and Theoretical Approach. ACS Applied Materials & Interfaces 2020.
  6. Green, D.W.; Willhite, G.P. Enhanced oil recovery; Henry L. Doherty Memorial Fund of AIME, Society of Petroleum Engineers …: 1998; Vol. 6.
  7. Escobar, E.; Valko, P.; Lee, W.; Rodriguez, M. Optimization methodology for cyclic steam injection with horizontal wells. In Proceedings of SPE/CIM International Conference on Horizontal Well Technology.
  8. Morte, M.K. Relative Permeability Determination for Steam Injection Processes: An Analytical Approach. 2016.”

Comment: Literature review is really poor and should be compared to the novelty of your work one by one. For example, say ...et al... did this and we present this. why is different and what are the novelties. It is not acceptable to say just the novel points without any comparison.

Response: We thank the reviewer for his/her comment, and we agree with it. The introduction section was improved as shown below:

“Some researchers have investigated different operating conditions in steam injection processes. Dong et al. [16] demonstrate that the main problems during steam injection were identified as steam breakthrough, low sweep efficiency and low steam efficiency. In addition, it is highlighted that the future of thermal processes lies in offshore fields. In addition, other types of technologies such as the use of electricity are currently under development (laboratory scale). The importance of studies that integrate EOR techniques at different scales and their effect through process simulation was also highlighted. Wen et al. [17] studied the interaction between heavy crude oil and water in the presence of the catalyst molybdenum oleate. This was done experimentally by replicating the aquathermolysis process in an autoclave, reducing crude oil quality 90 % at 240 °C. Furthermore, through a huff and puffin process at Liaohe oilfield, this process was carried out. In this process, higher crude oil productions were obtained, of up to 264.6 tons more, together with a reduction in the viscosity of the fluids of 78.2%. This work demonstrated the effect of the use of metals as catalysts to decrease the viscosity of crude oil together with high temperatures, leading to higher oil productions.”

“….. Some authors have included these nanoparticles to assist displacement tests simulating steam injection processes. Medina et al. [25] injected a dispersed nanofluid in a steam stream, containing 0.2% in mass fraction of CeO2 nanoparticles doped with 1.1% and 0.89% in mass fraction of Ni and Pd. They obtained 93.0% of oil recovery at 210 °C and X = 0.7. Franco et al. [21] used a nanofluid containing 0.05% in mass fraction of SiO2 nanoparticles functionalized with 1.0% in mass fraction of Ni and Pd. The nanofluid was injected in batch at 0.5 quality. In their work, water consumption was considerably reduced. The oil recovery was increased up to 50 %, and the crude oil quality was improved, reducing the asphaltene content and the oil viscosity. In another investigation, Afzal et al. [35] analyzed the catalytic effect at different concentrations of Fe2O3 nanoparticles on the viscosity of heavy oil at various temperatures. The treated crude oil had a viscosity of 16000cP and °API of 12. In this investigation, through displacement tests carried out on a laboratory scale, improvements in the relative viscosity of the crude oil of up to 60% were obtained considering steam injection and 0.5%wt of nanoparticles. In the displacement tests, an improvement in production from 38.31% to 68.41% was identified due to the use of nanomaterials. Finally, studies analyzing the effect of the catalyst size were carried out by Hamedi et al. [33] where nanometric nickel is compared with micrometric nickel in oil recovery and upgrading during steam injection. As a result, it is highlighted that the nanometric particles reduce the oil viscosity from 8500cP to 1530 cP, and oil recovery was increased 8.0%. This work highlights the importance of the catalysts, where the larger the size, higher the concentration of dispersant are needed to maintain stability, which affects the aquathermolysis process.

However, most of the studies reported in the literature that includes nanotechnology to assist the steam injection have been done at a fixed steam quality neither evaluated the effect of the quality. Therefore, the effect of nanoparticles during steam injection at different qualities for heavy oil recovery and upgrading is not yet clear.

In this context, the main objective of this study is to evaluate the effect of the steam quality in crude oil recovery and upgrading during steam injection in the presence of nanocatalysts.”

Comment: The conclusion section should be improved

Response: We thank the reviewes’ suggestion. The conclusions were modified as shown below:

  • The steam injection at 0.5 and 1.0 quality increased the oil recovery 3.0% and 7.0%, respectively, regarding the base curve (waterflooding). After nanofluid injection, the steam at 0.5 quality achieved an increase in oil recovery of 20.0% and 13.0%, for AlNi1 and AlNi1Pd1 nanoparticles, respectively. When steam was injected at 1.0 quality for both nanoparticles evaluated, no incremental oil was produced. The adsorption capacity was determined through batch adsorption experiments, obtaining higher asphaltene adsorption for the AlNi1Pd1 sample.
  • The catalytic experiments show that AlNi1 and AlNi1Pd1 samples achieve 100 % conversion of adsorbed asphaltenes at 220 °C during 140 and 175 min, respectively.
  • All functionalized nanoparticles tested in this work exhibit low energy activation compared with the virgin asphaltenes and a high gasification rate. The reaction mechanism depends on the metal's nature in the nanoparticles' surface and the interaction between different metals on the same surface.
  • The change in steam quality did not affect the crude oil upgrading (API and SARA content). However, AlNi1 and AlN1Pd1-based nanofluids increased de API gravity from 6.9° to 11°, and 13.3° and the asphaltene content was reduced near to 30% and 50%, respectively, and the highest viscosity reduction percentage was 99 % in for AlNi1Pd1.

Comment: The nomenclature section is missing and should be added

Response: We thank the reviewer for his/her recommendation. The nomenclature was added to the manuscript as shown below:

Nomenclature.

AlNi1

Alumina nanoparticles doped with 1.0% in mass fraction of Ni.

AlNi1Pd1

Alumina nanoparticles doped with 1.0% in mass fraction of Ni and Pd.

TEOR

Thermal Enhanced Oil Recovery

HO

Heavy oil

EHO

Extra-heavy oil

SBET (m2)

Surface area estimates by the Brunauer- Emmett-Teller surface method

EDX

Energy Dispersive X-ray spectroscopy

SEM

Scanning Electron Microscopy

DLS

Dynamic Light Scattering

Kabs

Absolute permeability

ko

Oil effective permeability

DVR

Degree of viscosity reduction

R%

Residue content

(cP)

reference viscosity

(cP)

viscosity after treatment

(J mol−1)

Change in standard Gibbs free energy

(J mol−1)

Change in standard enthalpy

(J (mol K)−1)

Change in standard entropy

 (mg g−1)

Henry's law constant

(g g−1)

Adsorbate association degree

 (g g−1)

Maximum adsorbed amount of adsorbate

 (mg L-1)

Equilibrium concentration

(mg L-1)

Initial concentration of asphaltene in the solution

  (J mol -1 K-1)

Universal gas constant

 (°C)

Absolute temperature

(kJ mol-1)

Activation energy

Function of reaction mechanism

X

Steam quality

Comment: Please add oil composition in the materials section

Response: We thank the reviewer for his/her comment. The crude oil composition was added to the materials section as shown next:

“The reservoir fluids are composed of an extra-heavy crude oil of 6.9° API, a viscosity of 2.3 ×106 cP at 25°C, and saturates, aromatics, resins and asphaltenes (SARA) content of 13.0%, 16.9%, 49.9%, and 20.2% in mass fraction, respectively.”

Reviewer 2 Report

The reviewer admires the amount of work presented by the author. Please consider the following comments to further improve the manuscript. 

Comments to author:

  1. The writing of this manuscript does not meet the standards of academic writing, which gets in the way of understanding the technical contents. The writing of the manuscript needs to be improved on the level of each sentence (wording, typo, grammar and sentence structure) by a professional editor. This should account for at least 50% of the revision efforts from the authors. 
  2. Revise the abstract to make it shorter and succinct. Focus on WHAT was done, HOW it was done, WHAT are the major findings and a significance statement.
  3. Include a nomenclature to document all symbols and abbreviations in the manuscript.
  4. Line 51, what does sub-statically mean? Is there a better word?
  5. Line 48, heat loss and poor heat conductance of rock.
  6. Line 65, delete and
  7. While referencing other works in literature, point out the improvement of oil recovery due to using nano fluid.
  8. Line 85, “In this work,” or “In their work”?
  9. Line 108, How can you possibly mobilize crude oil that is this viscous with water injection? What is the core temperature during water flooding? Please explain.
  10. Line 118, also contains
  11. In the description of experiments or methodology, it is highly recommended to use numbered steps. Make sure to point out all the materials used in each experiment and the external conditions. A table can be included to list all experiments done with materials and external conditions.
  12. Line 181, is it 3 or 6 stages?
  13. What is batch injection? Explain in text.
  14. For all figures, make sure to include explanations (name of equipment or abbreviations) of all equipment. For example, in the figure 1, what is the equipment between 12 and 13, what is the right most piece of equipment? Please explain under each figure.
  15. Line 201, do you mean inject water and oil at the same time? Again, the description of methodology is not clear and sufficient. Please include all important details of the experiment.
  16. This is trivial. For Table 1, list the first 3 properties for both media.
  17. Line 210, “The water flooding experiments the brine was injected until residual oil saturation (Sor) conditions are achieved.” What does this mean?
  18. I highly recommend separate section 2.4.1 into the Appendix, and only provide a summary of the results.
  19. Line 370. formatting issue.
  20. Review the name of EACH figure and table to make it succinct and clear. For example, Table 3, Properties of AlNi1 and AlNi1Pb1 nanoparticles.
  21. The result section in general is very difficult to read. I highly recommend reorganizing the result section. Carefully select the subtitles and discuss only ONE aspect or the effect of only ONE variable in one subsection. This makes the presentation clearer and more logical to read.
  22. How do you whether the decomposition contain volatile gas or not? If so, should it be considered in the effluent and how?
  23. Line 578, do you mean the residual conversion rate? How is it calculated and any reference?
  24. Line 595 to 597. Which specific data is used to make this conclusion?
  25. Use bullet points to list major findings.

Author Response

Medellín, May 20, 2021

Ms. Susan Ji

MDPI

RE: Response to reviewer’s comments regarding manuscript ID: processes-1233476

Dear Ms. Susan Ji

We would like to thank you for securing a prompt review of our manuscript titled; “Effect of Steam Quality on Extra-Heavy Crude Oil Upgrading and Oil Recovery Assisted with PdO and NiO-Functionalized Al2O3 Nanoparticles.” We have answered all the comments raised by the reviewer and have thoroughly revised the manuscript accordingly. We found the comments helpful and believe that our revised manuscript represents a significant improvement over our initial submission.

The detailed response (in blue) to the reviewer’s comments, suggestions, and questions (in black), and the revised manuscript are attached. As suggested, any track changes, highlights, or font colors in our revised manuscript have been removed and we believe now that our manuscript is publishable in Processes Journal.  

Please do not hesitate to contact us if you have any further questions.

Sincerely yours,

The authors

Reviewer 2.

Comment: The reviewer admires the amount of work presented by the author. Please consider the following comments to further improve the manuscript. 

Response: We thank the reviewer for his/her compliments on the importance of our study to the oil and gas industry and for his/her recommendation on publishing the manuscript in Processes Journal. This encourages us to keep the good work.

Comment: The writing of this manuscript does not meet the standards of academic writing, which gets in the way of understanding the technical contents. The writing of the manuscript needs to be improved on the level of each sentence (wording, typo, grammar and sentence structure) by a professional editor. This should account for at least 50% of the revision efforts from the authors. 

Response: We thank the reviewer for his/her comment. We apologize for the typos. A professional editor has revised the manuscript to improve the wording, grammar and sentence structures looking for the standards of academic writing.

Comment: Revise the abstract to make it shorter and succinct. Focus on WHAT was done, HOW it was done, WHAT are the major findings and a significance statement.

“This work focuses on evaluating the effect of the steam quality on the upgrading and recovery of extra-heavy crude oil in the presence and absence of two nanofluids. The nanofluids AlNi1 and AlNi1Pd1 consist of 500 mg·L-1 of alumina doped with 1.0% in mass fraction of Ni (AlNi1) and alumina doped with 1.0% in mass fraction of Ni and Pd (AlNi1Pd1), respectively, and 1000 mg·L-1 of tween 80 surfactant. Displacement tests were done in different stages including, i) basic characterization, ii) waterflooding, iii) steam injection at 0.5 quality, iv) steam injection at 1.0 quality, v) batch injection of nanofluids, vi) steam injection after nanofluid injection at 0.5 and 1.0 qualities. The steam injection was realized at 210 °C, the reservoir temperature was fixed at 80 °C, and pore and overburden pressure at 1.03 MPa (150 psi) and 5.51 MPa (800 psi), respectively. After the steam injection at 0.5 and 1.0 quality, oil recovery was increased 3.0% and 7.0%, respectively, regarding the waterflooding stage, and no appreciate upgrade in crude oil was observed. Then, during the steam injection with nanoparticles, the AlNi1 and AlNi1Pd1 increase the oil recovery in 20.0% and 13.0% at 0.5 steam quality, respectively. Meanwhile, when steam was injected at 1.0 quality for both nanoparticles evaluated, no incremental oil was produced. The crude oil was highly upgraded for AlNi1Pd1 system, reducing oil viscosity 99%, increasing API ° from 6.9 ° to 13.3°, and reducing asphaltene content 50% at 0.5 quality. It is expected that this work will eventually help to understand the appropriate conditions in which nanoparticles should be injected in a steam injection process to improve its efficiency in terms of oil recovery and crude oil quality.”

Comment: Include a nomenclature to document all symbols and abbreviations in the manuscript.

Response: We thank the reviewer for his/her recommendation. The nomenclature was added to the manuscript as shown below:

Nomenclature.

AlNi1

Alumina nanoparticles doped with 1.0% in mass fraction of Ni.

AlNi1Pd1

Alumina nanoparticles doped with 1.0% in mass fraction of Ni and Pd.

TEOR

Thermal Enhanced Oil Recovery

HO

Heavy oil

EHO

Extra-heavy oil

SBET (m2)

Surface area estimates by the Brunauer- Emmett-Teller surface method

EDX

Energy Dispersive X-ray spectroscopy

SEM

Scanning Electron Microscopy

DLS

Dynamic Light Scattering

Kabs

Absolute permeability

ko

Oil effective permeability

DVR

Degree of viscosity reduction

R%

Residue content

(cP)

reference viscosity

(cP)

viscosity after treatment

(J mol−1)

Change in standard Gibbs free energy

(J mol−1)

Change in standard enthalpy

(J (mol K)−1)

Change in standard entropy

 (mg g−1)

Henry's law constant

(g g−1)

Adsorbate association degree

 (g g−1)

Maximum adsorbed amount of adsorbate

 (mg L-1)

Equilibrium concentration

(mg L-1)

Initial concentration of asphaltene in the solution

  (J mol -1 K-1)

Universal gas constant

 (°C)

Absolute temperature

(kJ mol-1)

Activation energy

Function of reaction mechanism

X

Steam quality

Comment: Line 51, what does sub-statically mean? Is there a better word?

Line 48, heat loss and poor heat conductance of rock.

Line 65, delete and

Line 85, “In this work,” or “In their work”?

Line 118, also contains

Line 370. formatting issue.

Response: We thank the reviewer for his/her observation, and we apologize for the typos. The respective misprints were corrected in the revised manuscript.

Comment: While referencing other works in literature, point out the improvement of oil recovery due to using nano fluid.

Response: We thank the reviewer for his/her suggestion. The oil recovery obtained in the literature references were added as shown below:

“Medina et al. [21] injected a dispersed nanofluid in a steam stream, containing 0.2% in mass fraction of CeO2 nanoparticles doped with 1.1% and 0.89% in mass fraction of Ni and Pd. They obtained 93.0% of oil recovery at 210 °C and X = 0.7. Franco et al. [17] used a nanofluid containing 0.05% in mass fraction of SiO2 nanoparticles functionalized with 1.0% in mass fraction of Ni and Pd. The nanofluid was injected in batch at 0.5 quality. In their work, water consumption was considerably reduced. The oil recovery was increased up to 50 %, and the crude oil quality was improved, reducing the asphaltene content and the oil viscosity.”

Comment: Line 108, How can you possibly mobilize crude oil that is this viscous with water injection? What is the core temperature during water flooding? Please explain.

Response: We thank the reviewer for his/her comment. The following description was added to the manuscript to clear this part.

“Then, the water flooding experiments were done by introducing the brine from the water cylinder through the coil line into the porous media, until residual oil saturation (Sor) conditions are achieved. The water was injected at a flow ranging between 2.5 mL∙min-1 and 6.5 mL∙min-1. The temperature during this stage was the reservoir tem-perature (70 °C). This temperature is sufficient to promote the mobility of the crude oil in the porous medium according to our previous studies [43,44].”

Comment: In the description of experiments or methodology, it is highly recommended to use numbered steps. Make sure to point out all the materials used in each experiment and the external conditions. A table can be included to list all experiments done with materials and external conditions.

Response: We thank the reviewer for his/her suggestion. The methodology section was divided into different subsections to improve the understanding of the protocols. Also, details of the different materials used in each experiment were added to the text. The modified sections are enlisted below:

Section 2.2. Methods

Fig. 1 shows the proposed workflow to develop and evaluate two doped nanoparticles in crude oil upgrading and recovery in displacement tests varying the steam quality (0.5 and 1.0). The protocol starts with the synthesis and characterization of nanoparticles and their subsequent evaluation at static conditions through batch adsorption isotherms and thermogravimetric analysis (TGA). The TGA tests were done at isothermal and non-isothermal conditions. Then, coreflooding experiments were executed by injecting steam at different qualities (0.5 and 1.0) into the porous medium. The processes were assisted by two nanofluids (AlNi1 and AlNi1Pd1). Finally, the effluents recovered in each stage were robustly analyzed by API gravity, SARA distribution, and rheological behavior to determine the impact of steam quality and nanoparticles' chemical nature in crude oil upgrading.

Fig. 1. Workflow conducted in this investigation.

Section 2.2.2. Adsorption batch experiments

The nanoparticles' adsorption capacity was evaluated by batch adsorption experiments at 25 °C using asphaltene model solutions containing between 100 and 2000 mg·L-1 in toluene.”

Section 2.2.3. Catalytic decomposition of asphaltenes.

“The equipment was used to simulate the steam gasification of asphaltenes in the absence and presence of nanoparticles by injecting 100 mL·min-1 of N2 and 6.30 mL·min-1 of H2O(g) using a gas saturator filled with distilled water. The evaluation was done at non-isothermal condition in the temperature range of 100 °C- 600 °C at a heating rate of 10 °C·min-1 and isothermal heating using three temperatures (210 °C, 220 °C, and 230 °C).”

Comment: Line 181, is it 3 or 6 stages?

Response: We thank the reviewer for his/her observation, and we apologize for the typo. The displacement tests were done in six stages including i) basic characterization, ii) waterflooding, iii) steam injection at 0.5 quality (X = 0.5), iv) steam injection at 1.0 quality (X = 1.0), v) batch injection of nanofluids, vi) steam injection after nanofluid injection at 0.5 and 1.0 qualities.

Comment: What is batch injection? Explain in text.

Response: We thank the reviewer for his/her observation. According to the comment, the following was added to the Displacement test section.

“Batch injection refers to introducing a determined amount of liquid (nanofluid) into the porous medium at a defined injection rate. This technique does not use other agents such as gases for dispersion of the nanofluid. Fig 2. Shows a graphic representation of batch nanofluid injection.

Fig. 2. Graphical representation of nanofluid injection in batch into the porous medium.

Comment: For all figures, make sure to include explanations (name of equipment or abbreviations) of all equipment. For example, in the figure 1, what is the equipment between 12 and 13, what is the right most piece of equipment? Please explain under each figure.

Response: We thank the reviewer for his/her observation. The Figures were revised in detail. Below are shown the corrections done:

Fig. 3. Steam generation and displacement system for experimental tests. Legend: (1) positive displacement pumps, (2) oil-containing displacement cylinder, (3) brine-containing displacement cylinder, (4) water-containing cylinder, (5) nanofluid-containing cylinder, (6) tubular furnace, (7) manometers, (8) thermocouple, (9) pressure transducer, (10) slim tube, (11) sand packed bed, (12) sample output, and (13) hydraulic pump, (14) valves, (15) pressure multiplier. Reproduced from Medina et al. [24]. Copyright 2019, MDPI”

Comment: Line 201, do you mean inject water and oil at the same time? Again, the description of methodology is not clear and sufficient. Please include all important details of the experiment.

Response: We thank the reviewer for his/her comment. Details were added to the methodology as shown below:

“For absolute permeability estimation, 10 pore volumes (PV) of brine was injected into the porous media at a defined rate of 0.5 mLmin-1. Subsequently, the crude oil was injected until the pressure no longer changed and then 20 PV of water are injected for determining the water effective permeability (Kw) at residual oil saturation (Sor) conditions.”

“Then, the water flooding experiments were done by introducing the brine from the water cylinder through the coil line into the porous media, until residual oil saturation (Sor) conditions are achieved. The water was injected at a flow ranging between 2.5 mLmin-1 and 6.5 mLmin-1. The temperature during this stage was the reservoir temperature (70 °C).”

“…The incremental crude oil produced was estimated, and then steam was injected at X = 1.0 until there was no oil production. At this point, close to 15 and 16 PV of water equivalent to steam has been injected. Next, 0.5 PV of a hydrogen donor is placed in the porous media during a soaking time of 4 hours. Subsequently, 0.5 PV of the nanofluid is injected in batch at 0.5 mLmin-1 and left to act for 4 hours with the reservoir fluids. For both nanofluid presence scenarios, steam is again injected at X = 0.5 between 3 mL·min-1 and 5 mL·min-1 until no more oil is produced. Then, X was raised to 1 to verify the incremental oil production in the presence of nanoparticles

Comment: This is trivial. For Table 1, list the first 3 properties for both media.

Response: We thank the reviewer for his/her observation. According to the comment, Table 1 was corrected as follows:

Table 1. Petrophysics properties of porous media for the steam recovery processes in absence and presence of nanoparticles at X = 0.5 and X = 1.0 quality

System

Porous medium 1

Porous medium 2

Mineralogy

Silica 99%

Silica 99%

Length (cm)

60

60

Diameter (cm)

2.5

2.5

Porous volume (mL)

131.5

132

Porosity (%)

38

38

Absolute permeability

9080

9100

Nanoparticle injected (500 mg∙L-1)

AlNi1Pd1

AlNi1

Comment: Line 210, “The water flooding experiments the brine was injected until residual oil saturation (Sor) conditions are achieved.” What does this mean?

Response: We thank the reviewer for his/her comment. This expression was modifies as follows to clear its meaning:

“Then, the water flooding experiments were done by introducing the brine from the water cylinder through the coil line into the porous media, until residual oil saturation (Sor) conditions are achieved. The water was injected at a flow ranging between 2.5 mLmin-1 and 6.5 mLmin-1. The temperature during this stage was the reservoir temperature (70 °C).”

Comment: I highly recommend separate section 2.4.1 into the Appendix, and only provide a summary of the results.

Response: We thank the reviewer for his/her comment. Section 2.4.1 was changed as follows:

“Finally, the steam is injected at 210 °C and 1.90 MPa (276 psi) and 1.44 MPa (210 psi) to ensure a steam quality of 0.5 and 1.0, respectively. To guarantee the steam quality proposed in the experimental methodology, a model based on heat transfer equations and mass and energy conservation balances were made [45], identifying the change of steam through the generation and injection system. Details are found in section S1 of the Supplementary material information.”

Comment: Review the name of EACH figure and table to make it succinct and clear. For example, Table 3, Properties of AlNi1 and AlNi1Pb1 nanoparticles.

Response: We thank the reviewer for his/her observation. The name of the Figures and Tables was modified to make them succinct and clear, as shown below:

“Fig. 4. Energy-Dispersive X-ray images of AlNi1. (a) SEM image, (b) Al content, and (c) Ni content.

Fig. 5. Energy-Dispersive X-ray images of AlNi1Pd1. (a) SEM image, (b) Al content, (c) Ni content, and (d) Pd content.

Fig. 6. Rate of mass loss for non-catalyzed and catalyzed n-C7 asphaltene gasification by AlNi1 and AlNi1Pd1 nanoparticles.

Fig. 7. Isothermal conversion (α) for steam gasification of a) n-C7 asphaltenes in the absence (360 °C, 370 °C, and 380 °C) and presence (200 °C, 210 °C, and 220 °C) of b) AlNi1 and c) AlNi1Pd1 samples.

Fig. 8. Oil recovery curve profile for steam injection without and with AlNi1 and AlNi1Pd1 nanoparticles at different steam qualities (X=0.5 and X=1.0).

Fig. 9. API gravity for untreated extra heavy oil and crude oil recovered after the steam injection at X = 0.5, X = 1.0, and X = 0.5 + AlNi1 and AlNi1Pd1.

Fig. 10. SARA content for untreated extra heavy oil and crude oil recovered after the steam injection at X = 0.5, X = 1.0, and X = 0.5 + AlNi1 and AlNi1Pd1.

Fig. 11. Rheological behavior of recovered fluids at 25 °C. Virgin extra-heavy oil and oil recovered with steam injection at X = 0.5 and 1.0 and catalytic steam injection at X =0.5 assisted by AlNi1 and AlNi1Pd1 samples.

Table 2. Properties of nanoparticles AlNi1 and AlNi1Pd1 nanoparticles.

Table 3. Estimated effective activation energy ( ) and kinetic rate for isothermal catalytic steam gasification of asphaltenes in the absence and presence of AlNi1 and AlNi1Pd1 samples.

Table 4. Effect of steam injection at different qualities X=0.5 and X 1.0 and catalytic steam injection at X =0.5 in the presence of AlNi1 and AlNi1Pd1 in viscosity reduction degree (VRD) of an extra-heavy oil at 25 °C - shear rate of 10 s-1 and rheological parameters of the simulated model of Cross.”

Comment: The result section in general is very difficult to read. I highly recommend reorganizing the result section. Carefully select the subtitles and discuss only ONE aspect or the effect of only ONE variable in one subsection. This makes the presentation clearer and more logical to read.

Response: We thank the reviewer for his/her comment. The following subsections were included in the discussion and results section:

“3.3.1. Non-isothermal thermogravimetric experiments

3.3.2. Isothermal thermogravimetric experiments and kinetic analysis

3.4.1. Waterflooding and steam injection without nanoparticles at different qualities

3.4.2. Steam injection with nanoparticles at different qualities

3.5.1. API gravity changes

3.5.2. SARA composition changes

3.5.3. Simulated distillation

3.5.4. Rheological behavior analysis”

Comment: How do you whether the decomposition contain volatile gas or not? If so, should it be considered in the effluent and how?

Response: We thank the reviewer for his/her question. The analysis of the gaseous products generated during the different stages of the displacement test has been an important topic for the scientific communitythe scientific community and us. Given the complexity of this analysis, the experimental setup has been optimized and changed to obtain a reading of these compounds. So far, we have carried out some pilot tests in steam injection processes in the presence and absence of nanoparticles, mainly focused on producing a mixture rich in hydrogen. However, this is part of a future investigation that we hope to complete soon and share with the scientific community.

Comment: Line 578, do you mean the residual conversion rate? How is it calculated and any reference?

Response: We thank the reviewer for his/her observation. The residue conversion description was added to the section 2.5, as shown below:

“Simulated distillation (SimDis) was done to determine the residue conversion (R%). The residue content (620 °C+) was estimated using high-temperature simulated distillation (HTSD) following the ASTM D-7169 [48] in a gas chromatograph equipped with a capillary (Agilent, Needle 0.25 mm on the column, 5 ul syringe 3/PK). Results are given as residue conversion (R%) as Equation 2 shows:

(2)

   and , refer to the residue content before and after steam injection, respectively.”

Comment: Line 595 to 597. Which specific data is used to make this conclusion?

Response: We thank the reviewer for his/her observation. The following discussion was added to the manuscript to improve the sentence:

“Based on the behavior of viscosity as a function of the shear rate, it is observed that the virgin crude oil and the effluents obtained after the steam injection at 0.5 and 1.0 quality have a non-Newtonian behavior where the viscosity decreases by increasing the shear rate. However, the AlNi1Pd1 and AlNi1 samples could modify the crude oil rheological behavior from a pseudoplastic fluid to a Newtonian fluid, reducing the change in viscosity with increased shear rate.”

 Comment: Use bullet points to list major findings.

Response: We thank the reviewer for his/her suggestion. Conclusions were improved and enlisted as shown below: 

  • The steam injection at 0.5 and 1.0 quality increased the oil recovery 3.0% and 7.0%, respectively, regarding the base curve (waterflooding). After nanofluid injection, the steam at 0.5 quality achieved an increase in oil recovery of 20.0% and 13.0%, for AlNi1 and AlNi1Pd1 nanoparticles, respectively. When steam was injected at 1.0 quality for both nanoparticles evaluated, no incremental oil was produced. The adsorption capacity was determined through batch adsorption experiments, obtaining higher asphaltene adsorption for the AlNi1Pd1 sample.
  • The catalytic experiments show that AlNi1 and AlNi1Pd1 samples achieve 100 % conversion of adsorbed asphaltenes at 220 °C during 140 and 175 min, respectively.
  • All functionalized nanoparticles tested in this work exhibit low energy activation compared with the virgin asphaltenes and a high gasification rate. The reaction mechanism depends on the metal's nature in the nanoparticles' surface and the interaction between different metals on the same surface.
  • The change in steam quality did not affect the crude oil upgrading (API and SARA content). However, AlNi1 and AlN1Pd1-based nanofluids increased de API gravity from 6.9° to 11°, and 13.3° and the asphaltene content was reduced near to 30% and 50%, respectively, and the highest viscosity reduction percentage was 99 % in for AlNi1Pd1.

Round 2

Reviewer 2 Report

Most of the comments are well addressed. The writing needs further improvement. There are still many writing issues, for example, 

  1. Line 69, reducing the quality of crude oil by 90%... There are several other places that need to be changed as well. Please check carefully and revise thoroughly (line 106 etc.). 
  2. Use the built in equation editor in MS word to write the equations. MathType usually causes formatting issues when online proofing is used before publication. 
  3. The name of the last equipment is still not included in Fig. 3. 
  4. Line 272 to 273 can be revised to "Once the displacement test was complete, the produced fluids were characterized to evaluate the effect of nanoparticles on the the basic  properties of crude oil." It seems very unlikely that the manuscript has been proofread by a professional English editor. Please make sure proofreading is done thoroughly. 
  5. Equation 3, missing % 
  6. Line 387. Do you mean increase the conversion...?
  7. In Table 3, d_alpha and d_t are not included in the nomenclature. This column of data is barely discussed in the manuscript either.
  8. In Fig. 9. 38% increase in oil recovery 
  9. In the nomenclature, have you mentioned the Function of reaction mechanism in the body text? Include all and include only those symbols and abbreviations that appeared in the text. 
  10. It is highly recommended to use consistent formatting of figures that show data. 

Make sure the manuscript is proofread thoroughly and pay more attention to details in both formatting and writing. 

Author Response

Medellín, May 22, 2021

Ms. Susan Ji

MDPI

RE: Response to reviewer’s comments regarding manuscript ID: processes-1233476

Dear Ms. Susan Ji

We would like to thank you for securing a prompt review of our manuscript titled; “Effect of Steam Quality on Extra-Heavy Crude Oil Upgrading and Oil Recovery Assisted with PdO and NiO-Functionalized Al2O3 Nanoparticles.” We have answered all the comments raised by the reviewer and have thoroughly revised the manuscript accordingly. We found the comments helpful and believe that our revised manuscript represents a significant improvement over our initial submission.

The detailed response (in blue) to the reviewer’s comments, suggestions, and questions (in black), and the revised manuscript are attached. As suggested, any track changes, highlights, or font colors in our revised manuscript have been removed and we believe now that our manuscript is publishable in Processes Journal.  

Please do not hesitate to contact us if you have any further questions.

Sincerely yours,

The authors

Responses to the reviewer´s suggestions for the manuscript: Effect of Steam Quality on Extra-Heavy Crude Oil Upgrading and Oil Recovery Assisted with PdO and NiO-Functionalized Al2O3 Nanoparticles.

Reviewer 2.

Comment: Most of the comments are well addressed. The writing needs further improvement. There are still many writing issues, for example:

Line 69, reducing the quality of crude oil by 90%... There are several other places that need to be changed as well. Please check carefully and revise thoroughly (line 106 etc.). 

Response: We thank the reviewer for his/her effort made for securing a prompt review of our manuscript and for his/her suggestion. A professional editor has revised the manuscript to improve the wording, grammar and sentence structures looking for the standards of academic writing.

Comment: Use the built in equation editor in MS word to write the equations. MathType usually causes formatting issues when online proofing is used before publication. 

Response: We thank the reviewer for his/her suggestion. The MS word was used to write all the equations and symbols of the manuscript.

Comment: The name of the last equipment is still not included in Fig. 3. 

Response: We thank the reviewer for his/her observation. The Figure was revised in detail. Below are shown the corrections done:

“Fig. 3. Steam generation and displacement system for experimental tests. Legend: (1) positive displacement pumps, (2) oil-containing displacement cylinder, (3) brine-containing displacement cylinder, (4) water-containing cylinder, (5) nanofluid-containing cylinder, (6) tubular furnace, (7) manometers, (8) thermocouple, (9) pressure transducer, (10) slim tube, (11) sand packed bed, (12) sample output, and (13) hydraulic pump, (14) valves, (15) pressure multiplier and (16) gas col-lector. Reproduced from Medina et al. [24]. Copyright 2019, MDPI.”

Comment: Line 272 to 273 can be revised to "Once the displacement test was complete, the produced fluids were characterized to evaluate the effect of nanoparticles on the the basic  properties of crude oil." It seems very unlikely that the manuscript has been proofread by a professional English editor. Please make sure proofreading is done thoroughly. 

Response: We thank the reviewer for his/her observation. The manuscript was robustly revised. The sentence was changed as follows:

“The effluents obtained during the displacement test were characterized by API gravity, rheological behavior, SARA distribution, and residue content.”

Comment: Equation 3, missing % 

Response: We thank the reviewer for his/her appointment. The % symbol was added to the equation 3.

Plase, see the attachment

Comment: Line 387. Do you mean increase the conversion...?

Response: We thank the reviewer for his/her comment. We apologize for the typos. A professional editor has revised the manuscript to improve the wording, grammar and sentence structures looking for the standards of academic writing.

Comment: In Table 3, d_alpha and d_t are not included in the nomenclature. This column of data is barely discussed in the manuscript either.

Response: We thank the reviewer for his/her comment. The nomenclature was corrected as shown below:

Nomenclature.

AlNi1

Alumina nanoparticles doped with 1.0% in mass fraction of Ni.

AlNi1Pd1

Alumina nanoparticles doped with 1.0% in mass fraction of Ni and Pd.

TEOR

Thermal Enhanced Oil Recovery

HO

Heavy oil

EHO

Extra-heavy oil

SBET (m2)

Surface area estimates by the Brunauer- Emmett-Teller surface method

EDX

Energy Dispersive X-ray spectroscopy

SEM

Scanning Electron Microscopy

DLS

Dynamic Light Scattering

Kabs

Absolute permeability

ko

Oil effective permeability

DVR

Degree of viscosity reduction

R%

Residue content

(cP)

Reference viscosity

(cP)

Viscosity after treatment

 (cP)

Viscosity at the infinite shear rate

 (cP)

Viscosity at zero shear rate

H (mg g−1)

Henry's law constant

K (g g−1)

Adsorbate association degree

 (g g−1)

Maximum adsorbed amount of adsorbate

 (mg L-1)

Equilibrium concentration

 (mg L-1)

Initial concentration of asphaltene in the solution

 (°C)

Temperature

 (kJ mol-1)

Activation energy

X

Steam quality

Reaction rate

Additionally, the following discussion was added to the main text

“The AlNi1Pd1 presents a higher value for activation energy than AlNi1 due to the higher amount of molecules to decompose. Even though AlNi1Pd1 gives that result, the reaction still has the fastest reaction rate (dα/dt). This means that asphaltenes are cracked faster by AlNi1Pd1 than AlNi1, which is related to the low entropy produced due to the low molecular disorder of the adsorbed compounds (K) in AlN1Pd1. This result agrees with the adsorption affinity and catalytic activity through the thermogravimetric analysis [54].”

Comment: In Fig. 9. 38% increase in oil recovery 

Plase, see the attachment

Response: We thank the reviewer for his/her observation. The Figure was modified as shown next:

Comment: In the nomenclature, have you mentioned the Function of reaction mechanism in the body text? Include all and include only those symbols and abbreviations that appeared in the text. 

Response: We thank the reviewer for his/her comment, and we agree with it. The nomenclature was corrected using the symbols and abbreviations that appeared in the text, as follows:

Nomenclature.

AlNi1

Alumina nanoparticles doped with 1.0% in mass fraction of Ni.

AlNi1Pd1

Alumina nanoparticles doped with 1.0% in mass fraction of Ni and Pd.

TEOR

Thermal Enhanced Oil Recovery

HO

Heavy oil

EHO

Extra-heavy oil

SBET (m2)

Surface area estimates by the Brunauer- Emmett-Teller surface method

EDX

Energy Dispersive X-ray spectroscopy

SEM

Scanning Electron Microscopy

DLS

Dynamic Light Scattering

Kabs

Absolute permeability

ko

Oil effective permeability

DVR

Degree of viscosity reduction

R%

Residue content

(cP)

Reference viscosity

(cP)

Viscosity after treatment

 (cP)

Viscosity at the infinite shear rate

 (cP)

Viscosity at zero shear rate

H (mg g−1)

Henry's law constant

K (g g−1)

Adsorbate association degree

 (g g−1)

Maximum adsorbed amount of adsorbate

 (mg L-1)

Equilibrium concentration

 (mg L-1)

Initial concentration of asphaltene in the solution

 (°C)

Temperature

 (kJ mol-1)

Activation energy

X

Steam quality

Reaction rate

Comment: It is highly recommended to use consistent formatting of figures that show data. 

Response: We thank the reviewes’ suggestion. Figures were used in the same format as the reviewer suggested.

Comment: Make sure the manuscript is proofread thoroughly and pay more attention to details in both formatting and writing. 

.

Response: We thank the reviewer for his/her comment. We apologize for the typos. A professional editor has revised the manuscript to improve the wording, grammar and sentence structures looking for the standards of academic writing.
